Pseudocnidae of ribbon worms (Nemertea): ultrastructure, maturation, and functional morphology

http://orcid.org/0000-0003-4175-5007 Magarlamov Timur Yu 1 biotimur@yandex.ru
Turbeville James M. 2
Chernyshev Alexei V. 1
1 Far Eastern Branch, Russian Academy of Sciences, A.V. Zhirmunsky National Scientific Center of Marine Biology , Vladivostok , Russia
2 Department of Biology, Virginia Commonwealth University , Richmond, VA , USA
Magalhães Wagner
Electronic publication date: 2021 Feb 18
Publication date: 2021
Volume: 9
Electronic Location ID: e10912
Received 2020 Aug 31; Accepted 2021 Jan 16
Copyright: © 2021 Magarlamov et al.
Copyright year: 2021
Copyright holder: Magarlamov et al.
License: This is an open access article distributed under the terms of the Creative Commons Attribution License, which permits unrestricted use, distribution, reproduction and adaptation in any medium and for any purpose provided that it is properly attributed. For attribution, the original author(s), title, publication source (PeerJ) and either DOI or URL of the article must be cited.
License URL: https://creativecommons.org/licenses/by/4.0/

Keywords: Extrusomes, Pseudocnidae, Ultrastructure, Proboscis, Palaeonemertea, Pilidiophora

Funding: Russian Foundation for Basic Research 18-04-00704 Ministry of Science and Higher Education, Russian Federation 13.1902.21.0012 and 075-15-2020-796 The investigations were supported by the Russian Foundation for Basic Research grant (project No 18-04-00704) and the Ministry of Science and Higher Education, Russian Federation (grant ID 13.1902.21.0012, contract No 075-15-2020-796). The funders had no role in study design, data collection and analysis, decision to publish, or preparation of the manuscript.

==============================
The fine structure of mature pseudocnidae of 32 species of nemerteans, representatives of 20 genera, six families, and two classes was investigated with scanning and transmission electron microscopy. Pseudocnidae are composed of four layers (cortex, medulla, precore layer, and core) in most species investigated, but the degree of development and position of each layer can vary between different species. The secretion products comprising immature pseudocnidae segregate into separate layers: a thin envelope, which subsequently separates into the cortex and medulla and an extensive internal layer. We distinguish two pseudocnida types: type I is characterized by a two-layered core and type II by a three-layered core. Type I pseudocnidae are present in archinemertean species, Carinoma mutabilis, and in all pilidiophoran species, except Heteronemertea sp. 5DS; type II pseudocnidae occur in all studied species of Tubulanidae and the basal Heteronemertea sp. 5DS. Based on the structure of the discharged pseudocnidae observed in eleven species of palaeonemerteans and in eight species of pilidiophorans, we distinguish three different mechanisms (1–3) of core extrusion/discharge with the following characteristics and distribution: (1) the outer core layer is everted simultaneously with the tube-like layer and occurs in type I pseudocnidae of most species; (2) the extruded core is formed by both eversion of the outer core layer and medullar layer, and occurs in type I pseudocnidae of Micrura cf. bella; (3) the eversion of the outer core layer begins together with the core rod and core rod lamina and occurs in type II pseudocnidae. Morpho-functional comparison with other extrusomes (cnidae, sagittocysts, rhabdtites, and paracnids) confirm that pseudocnidae are homologous structures that are unique to nemerteans.

Introduction

Nemerteans are predominantly marine worms, currently comprising approximately 1300 species (Kajihara et al., 2008). Most species are carnivorous, using an eversible proboscis apparatus in prey capture (McDermott & Roe, 1985). The proboscis glandular epithelium of the “anoplan” nemerteans plays the primary role in the prey capture, and is positioned externally when the proboscis is everted, thus directly contacting the prey organism. This epithelium contains cells that secrete products that serve to immobilize the prey or effect its adhesion to the proboscis (Jennings & Gibson, 1969; Stricker & Cloney, 1983; Stricker, 1985; Kem, 1985). The glandular proboscis epithelium of Palaeonemertea and Pilidiophora with the exception Baseodiscus (Magarlamov & Chernyshev, 2011), Sonnenemertes (Chernyshev, Abukawa & Kajihara, 2015), and some Hubrechtella species (Kajihara, 2006) also contains specialized gland cells, which produce unique secretory granules with an internal hollow thread-like tubule (core) that are termed pseudocnidae.

The term pseudocnidae was introduced by Martin (1914), who considered the superficial similarity of these structures with cnidae of Cnidaria. Historically, pseudocnidae also have been referred to as nematocysts (Hubrecht, 1887), rhabdites (Gontcharoff, 1957; Hyman, 1959; Ling, 1971) or barbs (Jennings & Gibson, 1969). The pseudocnida core, which is analogous to the tubule of cnidarian cnidae, is capable of extrusion (Jennings & Gibson, 1969; Ling, 1971; Bürger, 1895). Jennings & Gibson (1969) mentioned that these structures can penetrate the integument of the prey, although evidence for this claim was not presented.

Most studies of pseudocnidae have been at the level of light microscopy, allowing only assessment of the shape, size, and location (Martin, 1914; Gerner, 1969; Gibson, 1985; Norenburg, 1993; Chernyshev et al., 2017). More recently, scanning electron microscopy (Chernyshev, Magarlamov & Turbeville, 2013; Magarlamov, Chernyshev & Turbeville, 2018) and confocal laser scanning microscopy studies contributed similar information (Magarlamov, Chernyshev & Turbeville, 2018). Ultrastructural studies of these secretory organelles have been limited to the following species: Lineus ruber-viridis (Gontcharoff, 1957; Ling, 1971; Anadón, 1976), Zygeupolia rubens (Coe, 1895), Tubulanus cf. pellucidus Coe, 1895 (Turbeville, 1991), Riseriellus occultus Rogers, Junoy, Gibson & Thorpe, 1993 (Montalvo et al., 1998), Cephalothrix cf. rufifrons (Johnston, 1837), Carinomella lactea Coe, 1905 (Turbeville, 2006), Hubrechtella juliae Chernyshev, 2003 (Chernyshev, Magarlamov & Turbeville, 2013), Balionemertes australiensis Sundberg, Gibson & Olsson, 2003, Cephalotrichella echinicola Chernyshev et al., 2018 and four Cephalothrix species (Magarlamov, Chernyshev & Turbeville, 2018). Turbeville (Turbeville, 1991; Turbeville, 2006) presented the first comparison of the pseudocnidae ultrastructure and identified three layers—an outer cortical layer (cortex), an inner medulla and a central tubular core. More recently, based on ultratstructural analyses of the pseudocnidae of B. australiensis, C. echinicola, and four species of Cephalothrix, we distinguished four layers (cortex, medulla, precore layer, and core), and this refinement enabled a more precise comparison of pseudocnida structure. However, the most accurate description of pseudocnida substructure will require ultrastructural examination of both mature pseudocnidae and their maturation process. Currently, data on maturation/formation are only available for H. juliae (Chernyshev, Magarlamov & Turbeville, 2013) and Cephalothrix simula Iwata, 1952 (Magarlamov, Chernyshev & Turbeville, 2018).

The size and shape of pseudocnidae have been used in species taxonomy of the genera Cephalothrix (Gerner, 1969) and Hubrechtella (Kajihara, 2006); however, Norenburg (Norenburg, 1993) pointed out that insufficient comparative analyses of pseudocnidae reduced their systematic value. Recently, we showed that pseudocnidae ultrastructural features could be used in the systematics of Archinemertea (Cephalotrichidae + Cephalotrichellidae) (Magarlamov, Chernyshev & Turbeville, 2018). This suggests a high potential of pseudocnida morphology for use in taxonomy of other nemerteans. The process of the core extrusion/discharge at the ultrastructural level has not been examined, making it difficult to determine the precise functions of pseudocnidae. An integrated approach to the study of pseudocnidae involving species from all major groups of Palaeonemertea and Pilidiophora promises a critical assessment of the utility of their morphology for nemertean systematics and should also provide insight into feeding biology of these animals.

We have undertaken a large-scale comparative analysis of pseudocnidae that includes data for 32 species representative of all major subсlades of Palaeonemertea and Pildiophora to allow for a comprehensive evaluation of structural diversity of pseudocnidae, to critically assess whether ultrastructural features will have implications for nemertean systematics, and to help clarify their function by elucidating the mechanisms of core extrusion. We also provide a morpho-functional comparison of pseudocnidae and cnidae of Cnidaria and extrusomes of other metazoans.

Materials and Methods

Specimen collection

The pseudocnidae of 32 nemertean species obtained from various regions and habitats of the Pacific and Atlantic oceans (see Table 1) were investigated. Live specimens were transported to the laboratory where most were identified to the species level. Some specimens could only be identified to the genus or family level (Table 1).

Table 1 Locality, number of specimens of the nemerteans examined.

Nemertean species	Locality	Number specimens	Comments to the species	
Class PALAEONEMERTEA	
Family Carinomidae	
Carinoma mutabilis Griffin, 1898	Oregon, USA	1		
Family Cephalotrichellidae	
Balionemertes australiensis Sundberg, Gibson & Olsson, 2003	Guam Island	2		
Cephalotrichella echinicola Chernyshev et al., 2018	Nha Trang Bay, Vietnam	2		
Family Cephalotrichidae	
Cephalothrix simula (Iwata, 1952)	Peter the Great Bay, Russia	4		
Cephalothrix hongkongiensis Sundberg, Gibson & Olsson, 2003	Yellow Sea, China	2		
Cephalothrix filiformis sensu Iwata, 1954	Peter the Great Bay, Russia	1	Undescribed species (Magarlamov, Chernyshev & Turbeville, 2018)	
Cephalothrix cf. rufifrons (Johnston, 1837)	St. Augustine, Florida, USA	1	Turbeville, 2006	
Family Tubulanidae	
Tubulanus punctatus (Takakura, 1898)	Peter the Great Bay, Russia	3		
Tubulanus cf. pellucidus (Coe, 1895)	Pawleys Island, South Carolina, USA	1		
Tubulanus sp. IZ 45559	Kuril Islands, Russia	1	Undescribed species (Kvist, Chernyshev & Giribet, 2015)	
Parahubrechtia sp.	Peter the Great Bay, Russia	2	Undescribed species (Kvist, Chernyshev & Giribet, 2015)	
Tubulanidae gen. sp. 33DS	Sea of Okhotsk, Russia	1	Undescribed species (Chernyshev & Polyakova, 2018)	
Tubulanidae sp. IZ 45557	Abyssal plain near Kuril–Kamchatka Trench		Undescribed species (Kvist, Chernyshev & Giribet, 2015)	
Tubulanidae sp. Kurambio2-17	Kuril-Kamchatka Trench, Russia	1	Undescribed species (Chernyshev & Polyakova, 2019)	
Class PILIDIOPHORA	
Family Hubrechtellidae	
Hubrechtella ijimai (Takakura, 1922)	East China Sea, South Korea	1		
Hubrechtella juliae Chernyshev, 2003	Peter the Great Bay, Russia	4		
Incertis sedia	
Heteronemertea sp.5DS	Okhotsk Sea, Russia	1	Undescribed basal species (Chernyshev & Polyakova, 2018)	
Family Lineidae	
Micrura ignea Schwartz and Norenburg, 2005	Caribbean Sea	1		
Nipponomicrura uchidai (Yamaoka, 1940)	Peter the Great Bay, Russia	2		
Nipponomicrura sp.	Peter the Great Bay, Russia	1	Undescribed species (Chernyshev, 2020)	
Micrura kulikovae Chernyshev, 1992	Peter the Great Bay, Russia	3		
Micrura cf. bella (Stimpson, 1857)	Sea of Japan, Russia	2	Chernyshev, 2020	
Micrura magna Yamaoka, 1940	Peter the Great Bay, Russia	1		
Notospermus albocinctus (Verrill, 1900)	Caribbean Sea	1	Transferred to Notospermus by Schwartz (Schwartz, 2009)	
Hinumanemertes kikuchii Iwata, 1970	Peter the Great Bay, Russia	1		
Maculaura aquilonia Hiebert and Maslakova, 2015	Taui Bay, Russia	1		
Maculaura sp.	Peter the Great Bay, Russia	1	Undescribed species (Chernyshev, 2020)	
Zygeupolia rubens (Coe, 1895)	Huntington Beach, South Carolina, USA	1		
Kulikovia alborostrata (Takakura, 1898)	Peter the Great Bay, Russia	3		
Cerebratulus cf. marginatus Renier, 1804	Peter the Great Bay, Russia	2	Undescribed species (Chernyshev, 2020)	
Lineus sanguineus (Rathke, 1799)	Sylt, Germany, North Sea	2		
Lineus viridis (Müller, 1774)	Sylt, Germany, North Sea	2		

Transmission and scanning electron microscopy investigations

Live worms were anesthetized using a 7% solution of magnesium chloride, and the proboscis was subsequently dissected from the body and cut into small fragments suitable for fixation. For transmission electron microscopy (TEM), proboscis fragments were fixed in 2.5% glutaraldehyde in phosphate-buffered saline (PBS) (pH 7.4) at room temperature and post-fixed with a 1% solution of osmium tetroxide in the PBS for 1 h. The fixed tissue was dehydrated in an ethyl alcohol and acetone series and embedded in Epon-Araldite or Polybed 812 resin. Alternatively, some specimens were fixed in 2.5% glutaraldehyde in Millonig’s phosphate buffer followed by postfixation in 1% osmium tetroxide in either Millonig’s phosphate or bicarbonate buffer. A Reichert Ultracut E Microtome or a Sorvall Porter-Blum MT-2b ultramicrotome were used to cut transverse and longitudinal thin sections (60–70 nm). Sections were stained with 1% uranyl acetate and 0.35% lead citrate solutions following standard protocols. Sections from 30 specimens were examined with a Zeiss Libra 120 transmission electron microscope while those of two specimens were examined with a Zeiss EM 9 S2 electron microscope.

For scanning electron microscopy (SEM) studies, the proboscides were anesthetized in 7% magnesium chloride and fixed in 2.5% glutaraldehyde in PBS (pH 7.4) at +4 °C and post-fixed with a 1% solution of osmium tetroxide for 1 h. The fixed material was dehydrated as described above, transferred to liquid CO2 and dried using the critical point drying machine Bal-tek EPD 030. Dried specimens were mounted with double-sided adhesive tape on aluminum SEM stubs, coated with platinum in a Q150T ion-sputter coater (Quorum), and examined under a Zeiss Evo 40 scanning electron microscope at 20 kV accelerating voltage.

For statistical analysis, measurements were obtained for 10 structures (n = 10).

Results

In this study, we adhere to the previously proposed four-layer organization of pseudocnidae (see Magarlamov, Chernyshev & Turbeville, 2018), but with two clarifications based on additional structural diversity revealed in this study: (1), We propose the use of the more common term—“core” instead of “filament-like core” or “filament” for the innermost layer of the pseudocnidae; and (2) The precore layer may be absent in pseudocnidae, and its location varies considerably (Fig. 1).

Figure 1 Schematic diagrams of pseudocnidae structure based on data herein and previously published observations.

(A) Carinoma mutabilis. (B) Balionemertes australiensis. (C) Cephalotrichella echinicola. (D) Cephalothrix simula. (E) Cephalothrix rufifrons. (F) Cephalothrix filiformis sensu Iwata (G) Tubulanus sp. IZ 45559. (H) Tubulanidae gen. sp. 33DS. (I) Tubulanidae sp. KuramBio2-17. (J) Tubulanidae sp. IZ 45557. (K) Tubulanus punctatus. (L) Tubulanus cf. pellucidus. (M) Parahubrechtia sp. (N) Hubrechtella juliae. (O) Hubrechtella ijimai. (P) Heteronemertea 5DS. (Q) Micrura ignea. (R) Nipponomicrura uchidai and Nipponomicrura sp. (S) Micrura cf. bella. (T) Micrura kulikovae. (U) Micrura magna. (V) Notospermus albocinctus. (W) Hinumanemertes kikuchii. (X) Maculaura aquilonia. (Y) Maculaura sp. (Z) Zygeupolia rubens. (AA) Kulikovia alborostrata. (BB) Cerebratulus cf. marginatus. (CC) Lineus sanguineus. (DD) Lineus viridis. (EE) Riseriellus occultus, the pseudocnida was reconstructed based on Montalvo et al. (1998). *Inner cortex layer—for pseudocnidae with bilaminar cortex, cortex—for pseudocnidae with unilaminar cortex; **Rod—for Tubulanidae and Heteronemertea gen. sp. 5DS, tubular structure—for other species.

Pseudocnidae are formed by secretory cells (pseudocnidae-forming cells) that are constituents of the proboscis glandular epithelium. Mature pseudocnidae of all species investigated are released from these pseudocnidae-forming cells onto the apical surface of the proboscis epithelium where they form monolayered clusters with the bottom of the layer oriented proximally (Figs. 2A–2J). In most nemerteans investigated the pseudocnidae are situated on the surface of bacillary secretory cells. Partially discharged apical granules of the bacillary cells form a sheet of material in which the mature pseudocnidae are embedded. Our investigations revealed that most pseudocnidae consist of four concentric layers of different electron density (Fig. 1).

Figure 2 Scanning electron micrographs of the everted proboscis.

(A) Carinoma mutabilis, panoramic view showing small (white asterisk) and large (black asterisk) pseudocnida clusters. (B) Cephalothrix hongkongiensis panoramic view showing pseudocnida clusters (arrowheads). (C) H. ijimai, spherical-shaped pseudocnidae with a slender protrusion on apical pole (arrowhead). (D) Hubrechtella juliae, panoramic view showing pseudocnidae with the extruded core (arrowheads). (E) Hinumanemertes kikuchii, panoramic view showing pseudocnida clusters (arrowheads). (F) Higher magnification of H. kikuchii, showing pseudocnida clusters (arrowheads). (G) Micrura kulikovae, pseudocnidae clusters. (H) M. kulikovae, panoramic view showing epithelial ridge (white arrowhead) along the proboscis surface (black arrowhead). (I) Cerabratulus cf. marginatus, panoramic view showing pseudocnida clusters (arrowheads). (J) Kulikovia alborostrata, panoramic view showing epithelial ridge (white arrowhead) along the proboscis surface (black arrowhead). (K) Higher magnification of a K. alborostrata, pseudocnida cluster.

Palaeonemertea

Carinoma mutabilis

In this species two types of mature pseudocnidae were distinguished based on size (Figs. 1A, 2A, 3A and 3B) and both exhibit an ovoid shape. Each type is localized in separate pseudocnid-forming cells. The pseudocnid-forming cells containing large pseudocnidae are more abundant (about 60–70%) than those with small pseudocnidae. Large pseudocnidae usually occur in clusters of 100 or more, whereas small pseudocnidae occur in clusters of 35–65 (Fig. 2A).

Figure 3 Transmission electron micrographs of the pseudocnidae of Carinoma mutabilis.

(A) Panoramic view showing small mature pseudocnidae released onto the apical surface of the proboscis epithelium. The apices face upward and the bases rest on secretions (asterisk) of underlying epithelial cells. (B) Panoramic view showing large mature pseudocnidae released on the proboscis epithelium surface. Pseudocnidae orientation is as in A. (C) Longitudinal section of a small pseudocnida. Arrowhead indicates the dense lamina of the cortex. (D) Transverse section of a small pseudocnida. Arrowhead points to the dense lamina of the cortex. (E) Longitudinal section of apical region of a small pseudocnida. (F) Longitudinal section of a large pseudocnida. The dense lamina of the outer cortex layer is indicated by the arrowhead. (G) Transverse section of a large pseudocnida. Arrowhead points to the dense lamina of the outer cortex layer. (H) Longitudinal section of the apical region of a large pseudocnida. Rectangle marks the contact of the precore layer with the inner cortex layer. (I) Longitudinal section of a pseudocnida with extruded core. Arrowhead points to the extruded core. Orientation as in A & B –ame, apical medulla; bme, basal medulla; co, cortex; cr, core; ico, inner cortex layer; me, medulla; ocl, outer core layer; oco, outer cortex layer; pc, precore layer; ts, tubular structure.

The middle region of small pseudocnidae measures 1.5–1.7 μm in length and 0.9–1.1 μm in diameter (Fig. 3C). The cortex consists of homogenous and moderately electron-lucent material (60–80 nm in thickness). The middle part of the cortex possesses a dense, thin lamina, which extends from the apex to the base of the pseudocnida (Figs. 3C and 3D). The medulla consists of a homogenous and moderately electron-dense matrix and is about 55 nm thick near apex and middle regions and up to 0.7 μm thick in the basal region of pseudocnida. The precore layer consists of homogenous and electron-dense material. The tubular core (approximately 0.7 μm long and 0.13–0.15 μm in diameter) extends from the apex of the pseudocnida and terminates in its midsection (Fig. 3C). The core merges with the cortex layer at the pseudocnida apex and both exhibit a similar electron density (Fig. 3E).

Large pseudocnidae are 2.3–2.7 μm in length and 1.2–1.4 μm in diameter (Figs. 3B, 3F and 3G). The cortex is differentiated into a moderately electron-lucent outer layer (70–120 nm in thickness) and a dense inner layer measuring 35 nm thick in the apical region and up to 120 nm thick in basal region (Figs. 3F and 3G). The dense lamina, located inside the outer cortex layer, extends from the apical end of the pseudocnida to its middle or basal region (Fig. 3F). The subcortical region is divided into two approximately equal parts (apical and basal) of different electron density. The apical half consists of a centrally situated core surrounded by a precore layer and medulla, whereas the basal half exhibits only the medulla (Fig. 3F). The core measures 0.8–1.1 μm in length and 0.25–0.35 μm in diameter and extends from the pseudocnida apex to either the midsection or base of the apical subcortical region. It consists of a tubular structure of moderate electron density (about 100 nm in thick) surrounded by a more electron-lucent outer core layer that is about 50 nm in thickness (Fig. 3H). The outer core layer merges with the outer cortex layer at the apex of the pseudocnida, and both have a similar electron density. The precore layer, measuring about 0.26 μm in thickness, consists of homogenous and highly electron-dense material (Figs. 3F and 3G). The precore layer merges with the inner cortex layer at the apex of the pseudocnida, and both exhibit similar electron density (Fig. 3H). The medulla, located in apical half of pseudocnidae, is moderately electron-dense and fills the area between the precore layer and inner cortex layer. The medulla is located in the basal half of the pseudocnida and is homogenous and moderately electron dense (Fig. 3F).

Some large pseudocnidae located within the apical surface of the proboscis epithelium have an extruded (=discharged) core (Fig. 3I). These discharged pseudocnidae are club-shaped and the extruded core extends parallel or at 3–5 degrees relative to the apical basal axis of pseudocnidae. The moderately electron-dense cortex of discharged pseudocnidae, measuring about 65 nm near the apical region of extruded core and up to 250 nm in basal, surrounds the medulla and its dispersed contents, that consist of densely packed, granular, highly electron-dense material embedded in a moderately electron-dense substance (Fig. 3I). The inner content of the discharged pseudocnida core consists of a dense granular matrix.

Balionemertes australiensis

The ultrastructure of the mature pseudocnidae of B. australiensis was previously examined (Magarlamov, Chernyshev & Turbeville, 2018) and in this study the stages of pseudocnida maturation are described and each layer the of pseudocnidae is specified (Fig. 1B).

The first step in development of a pseudocnida is the formation of a large, irregularly shaped granule as the elongate endoplasmic cisternae fuze (Fig. 4A). These granules exhibit a heterogenous composition, with the central portion characterized by a homogenous and highly electron-dense material, and a peripheral portion with homogenous and moderately electron-dense material. The early immature pseudocnidae are oval (Fig. 4B). At this stage, both a homogenous and moderately electron-dense cortex (~0.1 μm thick) and a medulla of similar density (0.1–0.35 μm thick) are apparent. The inner region of the granule (submedullar layer) is divided into unequal zones, a smaller apical- and larger basal zone (Fig. 4B). The apical zone consists of vesicles that are presumably cisternae of ER that exhibit an electron-lucent content. The basal zone is distinguished by its homogeneity and high electron-density. Externally, one large somewhat electron-lucent lateral process, comprised of expanded cisternae of the ER is attached to the middle part of immature pseudocnida (Figs. 4B and 4C). At the point of attachment of the lateral processes, the cortex is missing and the lateral processes directly contact the medulla. The outer surface of the lateral process is covered by a narrow (~30–50 nm thick) electron-dense layer, which merges with the cortex (Fig. 4C). In the following developmental stage, the pseudocnida is elongated along the apical-basal axis and has rounded ends (Fig. 4D). The medulla becomes thinner, and more electron-dense. The submedullar layer also exhibits changes, with the apical and basal zones now occupying approximately equal volume. The apical half contains two types of vesicles of a similar size range but distinguished from each other by their inner content; type I vesicles show electron-lucent content, the type II vesicles reveal highly electron-dense content (Fig. 4E). A tube-like structure or future core is formed in the central region of the apical half of a pseudocnida. The lateral processes are filled with small vesicles with electron-lucent content (Fig. 4D). Nearly mature pseudocnidae and mature pseudocnidae have a similar shape (rod with rounded ends) but are distinguished by the structure of the apical half of the submedullar region and the inner content of the lateral processes (For the structure of mature pseudocnidae see Figs. 4A–4D in (Magarlamov, Chernyshev & Turbeville, 2018) and Fig. 4G). The apical part of the submedullar region contains a centrally located core surrounded by a precore layer (Fig. 4F). The core consists of a centrally-situated tubular structure of moderate electron density (0.6–0.7 μm in thick) surrounded by a more electron-dense outer core layer measuring approximately 0.3 μm in thick. The precore layer is composed of a homogeneous and highly electron-dense material that measures up to 0.25 μm in thickness. The lateral process consists of homogeneous and moderately electron-dense material in which the residual vesicles of ER are distinguished (Fig. 4F).

Figure 4 Transmission electron micrographs of the pseudocnidae of Balionemertes australiensis (A–H), Cephalotrichella echinicola (I), Cephalothrix simula (J and K), Cephalothrix hongkongiensis (L and M), Cephalothrix cf. rufifrons (N and O), Cephalothrix filiformis sensu Iwata (P).

(A) Aggregation of cisternae of endoplasmic reticulum. (B) Longitudinal section of an early immature pseudocnida. Vesicles likely originating from ER cisternae, localized in lateral process (black arrow) and apical half of submedullar region (white arrow). (C) Attachment of lateral process (lp) to early immature pseudocnida. (D) Longitudinal section of an immature pseudocnida. (E) Longitudinal section of the apical half of an immature pseudocnida. (F) Longitudinal (center) and transverse (upper right) sections of nearly mature pseudocnidae. (G) Longitudinal section of a pseudocnida. Arrows point to apical extensions of the outer core layer. (H) High-magnification of a longitudinal section of a mature pseudocnida showing the cortex, medulla and precore layers with its narrow dense lamina (arrowhead). (I) Longitudinal section of a pseudocnida. (J) Longitudinal section of the apical region of a large pseudocnida. (K) Longitudinal section of apical region of a small pseudocnida. (L) Longitudinal section of the apical region of a large pseudocnida. (M) Longitudinal section of the apical region of a small pseudocnida. (N) Longitudinal section of the apical region of a large pseudocnida. (O) Longitudinal section of the apical region of a small pseudocnida. (P) Longitudinal section of apical region of a large pseudocnida. asm, apical submedullar layer; bsm, basal submedular layer; co, cortex; cr, core; dlp, dense lamina of lateral process; era, endoplasmic cisternae aggregate; lp, lateral process; me, medulla; ocl, outer core layer; pc, precore layer; ts, tubular structure.

The cortex and medulla of mature pseudocnidae are approximately similar in electron density, making it difficult to distinguish the two layers (Figs. 4G and 4H). The cortex is about 50 nm thick, whereas the thickness of the medulla varies from 0.1 to 0.3 μm in the apical and middle regions and reaches 0.9 μm in the basal region. The submedullar region is divided into two regions approximately equal in size (Fig. 4G). The apical half consists of a centrally situated core surrounded by a precore layer and the basal half exhibits only the precore layer. The precore layer has a narrow dense lamina on its outer surface (Fig. 4H). The core possesses a centrally-situated moderately electron dense tubular structure about 0.3 μm thick surrounded by an electron-dense outer core layer that is approximately 0.15 μm in thick. The outer core layer is apically expanded (Fig. 4G, arrows).

Cephalotrichella echinicola

Herein we refine the description of the layer arrangement of mature pseudocnidae of C. echinicola relative to previous work (Magarlamov, Chernyshev & Turbeville, 2018) (Fig. 1C). The cortex is about 60 nm thick and consists of homogenous and moderately electron-dense material (Fig. 4I), whereas the medulla is 0.1–0.3 μm thick and consists of homogenous and more electron-dense material. The submedullar (inner) region of the pseudocnidae is divided into two approximately equal parts (apical and basal). The apical half consists of a centrally situated filament-like core while the basal half exhibits only the precore layer (Fig. 4I). The core consists of a central tubular structure covered by an outer core layer.

Cephalothrix spp.

Descriptions of the core structures of previously evaluated pseudocnidae (Chernyshev, Magarlamov & Turbeville, 2013) of the following four Cephalothrix species were refined in this investigation: C. simula (Figs. 1D, 4J and 4K), C. hongkongiensis (Figs. 4L and 4M), C. cf. rufifrons (Figs. 1E, 4N and 4O), C. filiformis sensu Iwata (Figs. 1F and 4P). In all re-examined Cephalothrix species the core of both small and large pseudocnidae consists of a central tubular structure surrounded by an outer core layer. The outer core layer and cortex merge at the apex of the pseudocnida, and both layers have a similar electron density.

Tubulanus punctatus

Pseudocnidae are 2.0–2.4 μm in length and measure 0.85–1.1 μm in diameter at the base of the pseudocnida (Figs. 1G and 5A). They are ovate in form, possessing an expanded base and tapering distally to a slightly rounded apex. The cortex (about 80 nm thick) is homogenous and moderately electron-dense (Fig. 5B), whereas the medulla consists of a homogenous and highly electron-dense matrix and surrounds a core (Fig. 5B and 5C). There is no precore layer. The core, measuring 1.7–2 μm in length, 0.4–0.45 μm in diameter, extends from the apex of the pseudocnida and terminates about 0.4 μm from its base (Fig. 5B). The core consists of a central homogenous, electron-dense rod surrounded by two concentric layers of differing electron density (Fig. 5C). The outer core layer is about 80 nm thick and is comprised of a homogenous, moderately electron-dense matrix. The inner layer (core rod lamina) is thin and electron-dense (Figs. 5C–5E). The core rod lamina is separated from the rod and outer layer by an electron-lucent material. The basal region of the core exhibits only the outer core layer and core rod lamina (Fig. 5E). The outer core layer is connected to the inner surface of the cortex at the apical end of pseudocnida (Fig. 5D).

Figure 5 Transmission electron micrographs of the pseudocnidae of Tubulanus punctatus.

(A) Panoramic view showing mature pseudocnidae released on the apical surface of the proboscis epithelium. Pseudocnida apices are directed upward. (B) Longitudinal section of a pseudocnida. (C) Transverse section of a pseudocnida. High magnification inset shows core layers. (D) Longitudinal section of the apical region of a pseudocnida. (E) Longitudinal section of the basal region of a pseudocnida. (F) Panoramic view showing pseudocnidae with extruded cores (asterisks). (G) Scale-like structure of the cortex. (H) Transverse section of the extruded core of a pseudocnida. (I) Terminal end of extruded core with mushroom-shaped structure. (J) Longitudinal section of a portion of the extruded core of a pseudocnida. Arrows point to the connection of the inner layer of the extruded core with the pseudocnida cortex. co, cortex; cr, core; crl, core rod lamina; iec, inner layer of extruded core; me, medulla; ocl, outer core layer; oec, outer layer of extruded core; pc, precore layer; rd, core rod.

Some pseudocnidae situated on the apical surface of the proboscis epithelium exhibit an extruded core (Fig. 5F). The core (about 0.6 μm diameter) extends parallel to the apical basal axis of pseudocnida. The moderately electron-dense cortex of discharged pseudocnidae, measuring 110–180 nm in thickness, possesses hexagonal scale-like structures (Fig. 5G). The medulla region contains homogenous material of less electron density. The inner content of the discharged pseudocnida core contains the core rod covered by the core rod lamina (Fig. 5H). The terminal end of the discharged pseudocnida core possesses mushroom-shaped cap of densely packed, fine granules, which originated from core rod material (Fig. 5I). The wall of the extruded core is divided into an outer and inner layer, respectively measuring ~100 nm and ~30 nm in thickness (Fig. 5H). The inner layer of the extruded core is continuous with pseudocnida cortex (Fig. 5J).

Tubulanus cf. pellucidus

The pseudocnidae of this species are ovoid with an expanded base and a pointed apex (0.6–0.7 μm in length and 0.4–0.5 μm in diameter at their base) (Figs. 1L and 6A). The cortex measures approximately 30 nm in thickness and is homogenous and moderately electron-dense, whereas the medulla consists of homogenous and highly electron-dense material (Figs. 6B and 6C). The precore layer is 0.3–0.4 μm thick, homogenous and moderately electron-lucent. The core, measuring 0.55–0.65 μm in length, 0.2–0.3 μm in diameter, extends proximally from the apex of the pseudocnida and terminates about 0.15 μm from its base (Figs. 6A and 6B). The structure of the core of Tubulanus cf. pellucidus and T. punctatus is similar in organization, possessing a centrally-situated rod surrounded by a core rod lamina and a homogenous, moderately electron-dense outer core layer (Figs. 6B and 6C). Pseudocnidae with an extruded core were not observed for this species.

Figure 6 Transmission electron micrographs of the pseudocnidae of Tubulanus cf. pellucidus (A–C) and Tubulanus sp. IZ-45559 (D–H).

(A) Panoramic view showing mature pseudocnidae on the apical surface of the proboscis epithelium. The pseudocnida apices are directed towards the proboscis exterior and their bases rest on underlying gland cells (asterisk). (B) Longitudinal section of a pseudocnida. (C) Transverse section of a pseudocnida. (D) Panoramic view showing mature pseudocnidae on the proboscis epithelium surface. Orientation as in A. (E) Longitudinal section of pseudocnida. (F) Transverse section of pseudocnida. (G) Longitudinal section of the apical region of a pseudocnida. (H) Longitudinal section of the basal region of a pseudocnida. co, cortex; cr, core; crl, core rod lamina; me, medulla; ocl, outer core layer; pc, precore layer; rd, core rod.

Tubulanus sp. IZ-45559

Pseudocnidae of this species are similar in shape to those of T. punctatus and measure 2.8–3.4 μm in length and 1.1–1.4 μm in diameter at the base (Figs. 1G and 6D). The cortex (~0.1 μm thick) is homogenous and moderately electron-dense (Fig. 6E). The medulla measures up to 20 nm thick near the apex and up to 0.5 μm in its basal region. It consists of electron-dense concentric lamellae with a wavy profile separated from each other by moderately electron-dense, homogenous material (Figs. 6E and 6F). The precore layer is homogenous and moderately electron-lucent and is approximately 0.1 μm thick (Figs. 6F–6H). The tubular core measures 1.5–1.8 μm in length and 0.32–0.35 μm in diameter. It originates at the apex of the pseudocnidae and terminates about 0.5 μm from its base (Fig. 6E). The core structure of Tubulanus sp. and T. punctatus are similar consisting of a centrally-situated electron-dense rod covered by a core lamina and a broad outer core layer (Figs. 6F–6H). Pseudocnidae with an extruded core were not observed for this species.

Parahubrechtia sp.

Pseudocnidae of this species are rod shaped and measure 1.6–1.9 μm in length and about 0.47–0.56 μm in diameter (Figs. 1M and 7A) and occur in clusters of 300 or more per cell (Fig. 7B). In cross-sections the shape of pseudocnidae can vary from spherical to triangular (Fig. 7B). The cortex is about 50 nm thick and consists of homogenous and moderately electron-dense material (Figs. 7C–7E). The medulla is up to 50 nm thick near the apex of pseudocnida and up to 0.4 μm thick in the basal region of the pseudocnida. It consists of concentric electron-dense membrane-like lamellae extending parallel to the apical basal axis of pseudocnida (Figs. 7D–7F). The space between these lamellae is filled with more electron-lucent material. The precore layer is 10–50 nm thick and consists of electron-lucent homogenous material (Figs. 7C, 7E and 7F). The core is about 1.3 μm long and about 0.65 μm in diameter (Fig. 7C). The core morphology of Parahubrechtia sp. and T. punctatus is similar, distinguished only by the structure of the outer core layer. The outer core layer is 0.2 μm thick and possesses electron-dense borders enclosing somewhat electron-lucent material (Figs. 7D–7F).

Figure 7 Transmission electron micrographs of the pseudocnidae of Parahubrechtia sp.

(A) Panoramic view showing longitudinal sections of mature pseudocnidae on the proboscis epithelium surface. Apices of the pseudcnidae are oriented towards the top of the figure. The bases of the pseudocnidae rest on secretions of underlying gland cells (asterisk). (B) Transverse section through a proboscis papilla (= cluster of pseudocnidae). (C) Longitudinal sections of pseudocnidae. (D) Transverse section of a pseudocnida. (E) Longitudinal section of the apical region of a pseudocnida. (F) Longitudinal section of the middle region of a pseudocnida. (G) Panoramic view showing pseudocnidae with extruded cores. (H) Transverse section of middle region of a discharged pseudocnida. (I) Transverse section of the extruded core. co, cortex; cr, core; crl, core rod lamina; me, medulla; ocl, outer core layer; pc, precore layer; rd, core rod.

Some mature pseudocnidae with extruded cores are apparent on the apical surface of glandular epithelium of proboscis (Fig. 7G). The extruded core extends parallel to the apical-basal axis of pseudocnida. The moderate electron-dense cortex (approximately 90 nm thick) of discharged pseudocnidae, surrounds the medulla that contains a dense granular matrix with moderate electron-density (Fig. 7H). The inner region of the extruded core is filled with a homogenous matrix of high electron density (Fig. 7I). The external surface of the extruded core is covered by granular-fibrous material.

Tubulanidae gen. sp. 33DS

The pseudocnidae of this species are club-shaped and measure 1.95–2.35 μm in length and 0.6–0.7 μm in diameter at their base (Figs. 1H, 8A and 8B). The cortex, which measures about 40 nm in thickness, is homogenous and moderately electron-dense (Figs. 8B–8E) and surrounds a more electron-dense medulla. The medulla is about 65 nm thick near apex and up to 0.45 μm thick at the base of the pseudocnida (Fig. 8B). The homogenous, moderately electron-lucent precore layer is 30–60 nm thick. The core is 1.4–1.6 μm long and 0.25–0.31 μm in diameter. It extends from the apex of the pseudocnidae and terminates 0.4–0.7 μm from its base (Fig. 8B). The cores of Tubulanidae gen. sp. 33DS and Tubulanus punctatus are similar in organization, possessing a centrally-situated rod surrounded by the core rod lamina and a homogenous, moderately electron-dense outer core layer (Figs. 8C–8E). The core end located in the basal part of pseudocnida is about 0.35 μm long, is uniform and consists of densely-packed granules (Fig. 8E).

Figure 8 Transmission electron micrographs of the pseudocnidae of Tubulanidae gen. sp. 33DS.

(A) Panoramic view showing mature pseudocnidae released on the apical surface of the proboscis epithelium. Gland cells (ascterisks) underlie these structures. (B) Longitudinal section of a pseudocnida. (C) Transverse section of a pseudocnida. (D) Longitudinal section of the apical region of a pseudocnida. (E) Longitudinal section of the basal region of a pseudocnida. (F) Longitudinal section of a pseudocnida with an extruded core. (G) High magnification micrograph of a longitudinal section of the extruded core of a pseudocnida. (H) Transverse section of the extruded core of pseudocnidae (asterisks). co, cortex; cr, core; crl, core rod lamina; me, medulla; ocl, outer core layer; pc, precore layer; rd, core rod.

Mature pseudocnidae situated on the apical surface of proboscis epithelium exhibit discharged cores (Fig. 8F). The core extends at an angle of about 5 degrees relative to the apical basal axis of the pseudocnida. The electron-dense cortex of discharged pseudocnidae, measuring about 60 nm in thickness, surrounds the medulla and its dispersed contents. The inner content of the discharged pseudocnida core consists of loosely-packed material (Figs. 8G and 8H). The wall of the discharged pseudocnida core consists of apical and basal halves (Fig. 8F). The wall of the basal half, measures about 100 nm in thickness and is highly electron-dense, whereas the wall of the apical half measures about 50 nm thick and is moderately electron-dense (Fig. 8G).

Tubulanidae sp. IZ 45557

Pseudocnidae of this undescribed species are 3.7–4.0 μm in length and 0.7–0.8 μm in diameter (Figs. 1L and 9A). The cortex (~70 nm thick) is homogenous and moderately electron-lucent (Figs. 9B and 9C). The medulla is about 0.1 μm in thickness and is composed of a homogenous and moderately electron-dense matrix. A precore layer surrounds the core, where it is about 40 nm thick, and passes into the basal half of the pseudocnida below the core where it is about 270 nm thick (Fig. 9A). The region of the precore layer situated adjacent to the core is composed of a lightly stained homogenous material, while its basal (subcore) region consists of heterogeneous material, specifically irregularly-shaped darkly stained granules embedded in a more lightly stained homogenous matrix (Figs. 9A and 9B). The core, which measures about 2.2 μm in length and 0.30–0.4 μm in diameter, originates from the apex of the pseudocnidae and terminates about 1.0 μm from its base (Fig. 9A). The rod margin is electron-dense and surrounds a less dense central region resulting in a tubular morphology (Figs. 9B and 9C).

Figure 9 Transmission electron micrographs of the pseudocnidae of Tubulanidae sp. IZ 45557 (A–F) and Tubulanidae sp. Kurambio2-17 (G–K).

(A) Longitudinal (left) and transverse (right) sections of pseudocnidae. (B) Transverse sections of pseudocnidae. (C) Longitudinal section of the apical region of a pseudocnida. (D) Panoramic view showing longitudinally sectioned pseudocnidae with extruded cores. The bases of the pseudocnidae rest on the apical surface of the proboscis epithelium. Apices of pseudocnidae are oriented upward. (E) Longitudinal section of a pseudocnida with an extruded core (eco). (F) Transverse sections of extruded cores of pseudocnidae (arrowheads). (G) Longitudinal section of a pseudocnida. (H) Transverse section of a pseudocnida (arrowhead indicates a moderately electron-lucent homogenous matrix localized between the core rod lamina and the outer core layer). (I) Longitudinal section of the basal region of a pseudocnida (arrowhead indicates a moderately electron-lucent homogenous matrix localized between core rod lamina and the outer core layer). (J) Longitudinal section of the apical region of a pseudocnida. (K) Longitudinal section of a pseudocnida with the core extruded. High magnification inset shows the extruded core of the pseudocnida. The tip of the extruded core is oriented towards the top of the figure. co, cortex; cr, core; crl, core rod lamina; eco, extruded core; me, medulla; ocl, outer core layer; pc, precore layer; rd, core rod; ts, tubular structure.

A number of pseudocnidae reveal an extruded core (Fig. 9D). The core is oriented parallel or at a 3–5 degree angle with respect to the apical basal axis of pseudocnida (Fig. 9E). The discharged pseudocnidae exhibit an electron-dense cortex measuring 90–100 nm in thickness. The cortex surrounds the medulla region and consists of a loosely-packed fibrous matrix. The wall of the discharged pseudocnida core possesses a more electron-dense inner layer (about 35 nm thick) and moderately electron-dense outer layer (about 100 nm thick) (Figs. 9E and 9F). Its inner content consists of loosely-packed fibrous material. The external surface of the extruded core is associated with electron-dense granular material (Fig. 9F), which probably originated from granular material of the precore layer.

Tubulanidae sp. Kurambio2-17

Pseudocnidae of this tubulanid are club-shaped (Figs. 1M and 9G) and are 3.1–3.6 μm in length and 1.1–1.3 μm in diameter in the middle region. The homogenous and moderately electron-dense cortex is about 120 nm thick (Fig. 9H). The medulla is 0.1–0.2 μm thick near the apex region and up to 0.9 μm in the basal region of pseudocnida (Fig. 9G). It consists of dense, interconnected membrane-like lamellae with a wavy profile that are oriented parallel to the apical basal axis of pseudocnidae. These membranes are separated from each other by moderately electron-dense homogenous material (Fig. 9H). The precore layer is about 0.2 μm thick and is moderately electron-lucent. The precore layer in the basal region of pseudocnida is mushroom-shaped (about 0.8 μm in thickness) and comprised of an electron-lucent region filled by widely dispersed electron-dense granules of irregular shape (Figs. 9G and 9I). The core is 1.8–2.1 μm long and 0.28–0.45 μm in diameter. It extends from the apex and terminates in the middle half of the pseudocnida. It possesses a central, electron-dense rod (~0.1 μm in diameter) covered by a faintly discernible core lamina and a moderately electron-dense outer core layer measuring 50–100 nm in thickness. The core rod lamina and the outer core layer are separated from each other by a moderately electron-lucent homogenous matrix (Figs. 9H and 9I). The apical end of the core protrudes from pseudocnida apex (Fig. 9J).

Several mature pseudocnidae exhibit an extruded filament-like core (Fig. 9K). These discharged pseudocnidae are rod-shaped with the extruded core extended parallel to the apical basal axis of pseudocnida. The moderately electron-dense cortex of discharged pseudocnidae, measuring about 120 nm in thickness, surrounds the medulla region, and its dispersed contents consist of densely- or loosely-packed fibrous material. The inner content of the discharged pseudocnida core comprises a moderately electron-dense rod surrounded by electron-lucent homogenous material (Fig. 9K). The wall of the discharged pseudocnida core consists of two closely situated layers separated from each other by a moderately electron-lucent area (Fig. 9K). The external surface of the extruded core is covered by granular material.

Pilidiophora

Hubrechtella juliae

The ultrastructure and development of pseudocnidae of H. juliae were described in a previous study (Chernyshev, Magarlamov & Turbeville, 2013). In the current research, the naming of the layers and the core structure of mature pseudocnidae were refined (Fig. 1N). Pseudocnidae are covered by a cortex that has a bilaminar organization: the outer layer consists of close-packed fibrous material of moderate electron-density, and the inner layer is composed of homogenous material of higher electron density (Fig. 10A, inset). The acentrically located core consists of tubular structure surrounded by outer core layer (Fig. 10B). The core extends from the apex to the cortex at the base of the pseudocnida (Fig. 10C). Immediately outside the core is a surrounding precore layer (Figs. 10A and 10B). The precore layer consists of loosely-packed fibrous material of moderate electron density. The area between the precore layer and the cortex is encompassed by a medulla, which consists of spherical granules measuring about 0.2–0.3 μm (Fig. 10A). The thin peripheral region of the medulla is free of granules.

Figure 10 Transmission electron micrographs of the pseudocnidae of Hubrechtella juliae (A–E) and Hubrechtella ijimai (F–J).

(A) Transverse section of a pseudocnida. High maginification inset reveals the bilaminar cortex. (B) Transverse section of a pseudocnida core. (C) Longitudinal section of the basal region of a pseudocnida. (D) Longitudinal section of a pseudocnida with an extruded core. High magnification inset shows the extruded core of the pseudocnida. (E) Transverse section of discharged pseudocnida. (F) Panoramic view showing longitudinal sections of two mature pseudocnidae on the apical surface of the proboscis epithelium. Apices of the pseudocnidae are directed upward and the core is visible in the center of each. (G) Transverse section of pseudocnida. (H) Transverse section of a pseudocnida core. (I) Longitudinal section of apical region of a pseudocnida. (J) Longitudinal section of basal region of the pseudocnida. cr, core; ico, inner cortex layer; me, medulla; ocl, outer core layer; oco, outer cortex layer; pc, precore layer; ts, tubular structure.

Some pseudocnidae situated on the apical surface of the proboscis epithelium have an extruded core (Fig. 10D). These discharged pseudocnidae are rod-shaped and the extruded core extends parallel or at 5–15 degrees relative to the apical basal axis of pseudocnidae. The cortex of discharged pseudocnidae is not bilaminar and consists of material of moderate electron-density, measuring about 0.5 μm in thickness (Fig. 10E). The medulla contains homogenous material of lower electron density, and transverse sections of the medulla reveal folded inner margins (Fig. 10E). The central region of extruded pseudocnidae contains loosely-packed fibrous material. The wall of the extruded core consists of a thin, dense inner layer that is covered by densely-packed fibrous material (Fig. 10D, inset). The inner content of the discharged pseudocnida core consists of loosely- or moderately-packed fine, granular material.

Hubrechtella ijimai

Mature pseudocnidae of H. ijimai are released onto the epithelial surface in small groups of 2–4 or individually (Figs. 1O, 2C and 10F). They are spherically-shaped and about 4.1–5.3 μm in diameter with a small protrusion on the apical pole (Fig. 2D). The cortex is differentiated into a moderately electron-dense outer layer measuring 0.1–0.2 μm in thickness and a dense inner layer that is 0.1–0.15 μm in thick (Fig. 10G). The medulla fills the central region of pseudocnida and is moderately electron-dense and homogenous in composition (Fig. 10G). The core is 3.9–4.8 μm long and 0.45–0.55 μm in diameter and extends from the apex to the inner cortex layer at the base of the pseudocnida (Figs. 10F, 10I and 10J). It consists of tubular structure that is about 0.34 μm thick surrounded by the thinner (~ 0.14 μm thick) outer core layer (Figs. 10I and 10J). The tubular structure possesses an electron-dense wall enclosing less dense homogenous content. The outer core layer consists of 4–7 highly electron-dense concentric lamellae each separated by electron-lucent material (Fig. 10H). The lamellae are orientated parallel to the apical-basal axis of pseudocnidae. The outer core layer is surrounded by the precore layer that consists of highly electron dense material (~50 nm thick). The precore layer merges with the inner cortex layer at the pseudocnida apex and both exhibit a similar electron density (Fig. 10I). Pseudocnidae with an extruded core were not observed for this species.

Heteronemertea gen. sp. 5DS

Pseudocnidae of this undescribed hereronemertean are 2.0–2.3 μm in length and about 0.8–0.9 μm in diameter (Figs. 1P and 11A). The cortex is bilaminar, consisting of, a fibrous moderately electron-lucent outer layer approximately 50–60 nm thick and an inner layer of homogenous material of greater electron density measuring approximately 30–40 nm in thickness (Figs. 11B and 11C). The medulla consists of an electron-dense layer measuring about 30 nm in thickness near the apex and middle regions of pseudocnida and up to 0.2 μm thick in the basal region of pseudocnida (Fig. 11A). The precore layer consists of moderately dense material forming a maze-like pattern within a lucent homogenous matrix (Figs. 11A–11C). The dense material of the precore forms a ring around the core with short branches directed towards the medulla of the pseudocnida. The thin region of the precore layer adjacent to the core is free from dense material (Figs. 11A–11C). The core is 1.9–2.2 μm long and 0.13–0.14 μm in diameter and extends from the pseudocnida apex towards its base, terminating near the border of the medulla and cortex (Fig. 11A). It possesses a highly electron-dense rod measuring about 45 nm in diameter, a surrounding thin lamina and outer core layer that is about 40 nm in thick and exhibits moderate electron density and a homogenous consistency (Fig. 11C). Some pseudocnidae possess a partially everted outer core layer with the core rod protruded on pseudocnida apex (Fig. 11D).

Figure 11 Transmission electron micrographs of the pseudocnidae of Heteronemertea sp. 5DS.

(A) Longitudinal section of a pseudocnida. (B) Transverse section of a pseudocnida. (C) Longitudinal section of the apical region of a pseudocnida. (D) A pseudocnida with partially everted core. (E) Longitudinal section of a pseudocnida with an extruded core (asterisk). (F) Longitudinal section of a portion of an extruded core of a pseudocnida. (G) Transverse section of the extruded core of a pseudocnida. (H) Immature pseudocnida at early maturation stage. (I) Immature pseudocnida at the late maturation stage. (J) Nearly mature pseudocnida with partially formed medulla. (K) Nearly mature pseudocnida with medulla containing formed lamellae. cr, core; crl, core rod lamina; ico, inner cortex layer; ls, lamellar structure; me, medulla; pc, precore layer; ocl, outer core layer; oco, outer cortex layer; rd, core rod.

A few mature pseudocnidae show an extruded core (Fig. 11E). The cortex and medulla of discharged pseudocnida retain a structure similar to that of the undischarged pseudocnida. The inner content of discharged pseudocnidae consist of closely-packed fibrous material, whereas the inner content of the discharged core possesses a centrally-positioned rod covered by loosely-packed fibrous material (Figs. 11F and 11G). The wall of the extruded core is about 40 nm thick and continuous with pseudocnida cortex; both exhibit a similar density. The external surface of the extruded core is covered by fibrous material (Fig. 11G).

Pseudocnidae in different stages of development were also examined. The initial stage in pseudocnida maturation is formation of a large rounded secretory granule measuring 0.65–0.8 μm in diameter (Fig. 11H). The granule consists of a moderately electron-dense homogenous wall (~0.25 μm thick) surrounding electron dense, homogenous material. The immature pseudocnidae at the early maturation stage are ovoid and are up to 1.8 μm in length and 0.9–1.1 μm in width (Fig. 11I). Their wall consists of closely-packed fibrous material with moderate electron density enclosing more loosely packed fibers. The immature pseudocnidae at the late maturation stage are oval-shaped and measure up to 1.8 μm in length and 0.9–1.1 μm in width (Fig. 11J). Their wall consists of closely-packed fibrous material with moderate electron density enclosing more loosely packed fibers. The core is not found at this stage. The nearly mature pseudocnidae are round or oval in shape and reach a length of 1.3 μm and a width of 0.8–1.1 μm (Fig. 11K). The cortex is 30–170 nm thick and consists of closely-packed moderately electron-dense fibrous material. The medulla is about 60 nm thick and is composed of a highly electron-dense homogenous material. The inner surface of medulla exhibits lamellae (Fig. 11K). A rudimentary core measuring about 0.7 μm long and about 0.1 μm in diameter is located at the apex of pseudocnida and consists of a central rod surrounded by a dense lamina (Figs. 11J and 11K).

Micrura ignea

The pseudocnidae are rod-shaped measuring 4.5–5.5 μm in length and about 1.1–1.4 μm in diameter (Figs. 1Q and 12A). The cortex is about 50–70 nm thick and consists of a homogenous and moderately electron-lucent inner layer covered by a thin and electron-dense outer layer (Fig. 12B). The medulla consists of homogenous and moderately electron-dense material and is about 0.2 μm thick near apex and middle regions and up to 0.9 μm thick in the basal region of pseudocnida (Fig. 12A). In the basal region of the pseudocnida the medulla frequently contains as many as 10 round vesicles of moderate electron-lucency. In this species the submedullar (inner) content of the pseudocnidae is partitioned into two approximately equal regions (apical and basal), which are separated from each other by sphincter-like structure (Fig. 12A). The wall of the sphincter-like structure possesses a honey-comb-like formation with electron-dense borders enclosing more lucent material (Fig. 12A, inset). The apical half of the submedullar region consists of a centrally situated core (Figs. 12A and 12B). The core, measuring 1.8–2.1 μm in length and 0.45–0.55 μm in diameter, extends from the pseudocnida apex towards its base, terminating at the boundary of the anterior and posterior submedullar regions. The core consists of a tubular structure of moderate electron density about 0.1 μm in thickness surrounded by a more electron-lucent outer core layer that is about 0.3 μm in thick. The middle half of the tubular structure is surrounded by a thin cylinder of dense material (Figs. 12B and 12C). The outer core layer and inner layer of the cortex merge at the apex of the pseudocnida, and both layers have a similar electron density (Fig. 12B). The basal submedullar region of the pseudocnida exhibits only the precore layer, which consists of homogenous and moderately electron-lucent material (Fig. 12A). Pseudocnidae with an extruded core were not observed for this species.

Figure 12 Transmission electron micrographs of the pseudocnidae of Micrura ignea.

(A) Longitudinal section of pseudocnidae. High magnification inset shows a honeycomb-like sphincter (arrow). (B) Longitudinal section of apical region of a pseudocnida revealing a thin ring of dense material (arrow) on outer surface of the tube-like structure (tl). (C) Transverse section of a pseudocnida. The tube-like structure (tl) is covered with a thin cylinder of dense material (arrow). cr, core; me, medulla; ico, inner cortex layer; ocl, outer core layer; oco, outer cortex layer; pc, precore layer; sls, sphincter-like structure.

Nipponomicrura uchidai

Pseudocnidae are somewhat club-shaped, and measure 3.1–3.7 μm in length and 1.2–1.4 μm in diameter at the expanded basal portion of the pseudocnida (Figs. 1R and 13A). The cortex is bilaminar; the outermost or peripheral layer is composed of electron-dense material and is 20 nm thick, whereas the inner layer consists of moderately electron-lucent homogenous material and is 40 nm thick (Figs. 13B and 13C). The medulla consists of moderately electron-dense homogenous material and is about 0.16 μm thick near the apex and middle regions of pseudocnida and up to 0.8 μm thick in the basal region of pseudocnida (Fig. 13A). The medulla possesses 2–3 dense concentric lamellae oriented parallel to the apical basal axis of pseudocnida (Figs. 13A and 13C). There is no precore layer between the medulla and core. The core, measuring 2.4–2.7 μm in length and up to 0.5 μm in diameter, extends from the apex and terminates about 0.7 μm from the base of the pseudocnida (Fig. 13A). The core consists of concentric layers of differing electron densities: a centrally-situated tubular structure of electron-dense material, a lamina of greater density and a moderately electron-dense homogenous outer core layer (Figs. 13B and 13C). The tubular structure and lamina are divided by moderately electron-lucent homogenous matrix. The basal and middle half of outer core layer is surrounded by thin tubular structure consisting of electron-dense material, the supportive core layer (Figs. 13A and 13C).

Figure 13 Transmission electron micrographs of the pseudocnidae of Nipponomicrura uchidai (A–E) and Nipponomicrura sp. (F–J).

(A) Longitudinal section of a pseudocnida with the core surrounded by a thin ring of dense material (supportive core layer). (B) Transverse section of apical region of the pseudocnida. (C) Transverse section of middle region of the pseudocnida. (D) Longitudinal section of a pseudocnida with a discharged core. The core apex is at the top of the figure. High magnification inset shows the wall of pseudocnida, which is the outer trilaminar structure bordering the homogenous core. (E) Transverse section of the discharged core of a pseudocnidae. (F) Longitudinal section of two pseudocnidae. (G) Transverse section of the middle region of pseudocnidae showing the core surrounded by a thin ring of dense material (supportive core layer). (H) Longitudinal section of pseudocnidae with extruded cores (asterisks). The core apices are directed upwards. (I) Longitudinal section of the discharged core and its connection with the pseudocnida body (arrows). (J) Longitudinal section of the discharged core of a pseudocnida. co, cortex; cr, core; crl, core rod lamina; ico, inner cortex layer; lm, laminar structure; me, medulla; ocl, outer core layer; oco, outer cortex layer; sup, supportive core layer; ts, tubular structure.

Some mature pseudocnidae localized on the apical surface of the glandular proboscis epithelium have an extruded core (Fig. 13D). These discharged pseudocnidae are club-shaped with the extruded core oriented parallel to the apical-basal axis of pseudocnidae. The discharged pseudocnidae possess a three-layer wall: a fibrous densely-packed moderately electron-dense outer layer (about 40 nm in thick), a lamellar and moderately electron-dense middle layer about 45 nm thick and a highly electron-dense homogenous inner layer that is about 20 nm in thick (Fig. 13D). The wall of discharged pseudocnidae surrounds the medulla, which corresponds in density to the medulla of the undischarged pseudocnida. The inner content of the discharged pseudocnida core consists of moderately electron-dense homogenous material (Figs. 13D and 13E), and its external surface is covered by densely packed fibrous material.

Nipponomicrura sp.

The ultrastructure of pseudocnidae of N. uchidai and Nipponomicrura sp. is distinguished only by medulla structure. The medulla in Nipponomicrura sp. consists of moderately electron-dense homogenous material without lamellar structures (Figs. 1R, 13F and 13J).

Some pseudocnidae localized on the apical surface of the proboscis epithelium have an extruded core (Figs. 13H–13J). The ultrastructure of discharged pseudocnidae of N. uchidai and Nipponomicrura sp. is distinguished by the wall structure and core inner content (Figs. 13D and 13J). The wall of discharged pseudocnidae of Nipponomicrura sp. is homogenous and moderately electron dense. The inner content of the discharged pseudocnida consists of densely packed material surrounded by moderately electron-dense homogenous material (Fig. 13J).

Micrura kulikovae

Pseudocnidae of this species are rod-shaped, measuring 3.0–3.5 μm in length and 0.7–0.9 μm in diameter (Figs. 1S, 2F, 14A and 14B). The pseudocnidae form clusters of 120–140 per cell, which rest on the apical surface of bacillary gland cell extensions (Figs. 1F and 14A). The pseudocnidae clusters are localized in an epithelial ridge along the proboscis surface (Fig. 2H). Their bilaminar cortex consists of an electron-dense homogenous outer layer (~20 nm in thickness) and a moderately electron-lucent homogenous inner layer (~40 nm in thickness) (Figs. 14C and 14D, inset). The medulla, measuring approximately 0.15–0.25 μm in thickness near the apex and middle regions of pseudocnida and up to 0.6 μm thick in the basal region of pseudocnida, is composed of moderately electron-dense homogenous material (Fig. 14B). The submedullar (inner) content of the pseudocnidae is divided into two approximately equally sized apical and basal regions (Fig. 14B). The apical half consists of a centrally situated filament-like core measuring 1.1–1.5 μm in length and 0.22–0.24 μm in diameter. The core extends from the apex to the middle region of the pseudocnida (Fig. 14B) and consists of two concentric layers with different electron density (Fig. 14D). The inner layer forms a tubular structure of moderate electron density and is surrounded by an outer homogenous layer of higher electron density. At the pseudocnida apex, the outer core layer terminates on the inner surface of the cortex, whereas the inner core layer extends to the pseudocnida surface (Fig. 14C). The basal half of the submedullar region of the pseudocnida exhibits only the precore layer, which is composed of moderately electron-dense irregularly-shaped granules within moderately electron-lucent homogenous material (Fig. 14B). Pseudocnidae with an extruded core were not observed for this species.

Figure 14 Transmission electron micrographs of the pseudocnidae of Micrura kulikovae (A–D) and Micrura cf. bella (E–J).

(A) Panoramic view showing mature pseudocnidae on the apical surface of the proboscis epithelium. The apical ends are directed upward and the bases overlie secretions of underlying gland cells (asterisk) of the epithelium. (B) Longitudinal section of pseudocnidae. (C) Longitudinal section of the apical region of a pseudocnida. (D) Transverse section of a pseudocnida. High magnification inset shows the wall of a pseudocnida. (E) Longitudinal sections of pseudocnidae. (F) Transverse section of a pseudocnida. (G) Longitudinal section of the apical region of a pseudocnida. (H) Longitudinal sections of pseudocnidae with extruded cores. (I) Longitudinal sections of the proximal part of the extruded core (ecr). (J) Terminal end of the extruded core revealing a mushroom-shaped structure. co, cortex; cr, core; ico, inner cortex layer; me, medulla; ocl, outer core layer; oco, outer cortex layer; ts, tubular structure.

Micrura cf. bella

Pseudocnidae are rod-shaped measuring 3.5–4.5 μm in length and about 1.0–1.2 μm in diameter (Figs. 1S and 14E). The cortex is about 0.1 μm thick and is divided into two parts; a thin, electron-dense outer layer and a homogenous, moderately electron-dense inner layer (Figs. 14F and 14G). The medulla is about 0.2 μm thick near apex and middle regions of pseudocnida and up to 0.7 μm thick in the basal region. Its contents are homogenous and moderately electron dense (Fig. 14F). The submedullar (inner) region of the pseudocnidae is divided into two approximately equal parts (apical and basal). The apical half consists of a centrally situated filament-like core. The core, measuring 1.2–1.4 μm in length and 0.38–0.41 μm in diameter, extends from the pseudocnida apex towards its base, terminating at the boundary of the apical and basal submedullar regions (Fig. 14E). The core consists of concentric layers of differing electron densities: a centrally-situated tubular structure of moderate electron density and a homogenous outer core layer exhibiting somewhat greater electron density. At the pseudocnida apex, the outer core layer merges with the cortex and both have a similar electron density (Fig. 14G). The submedullar region exhibits only the precore layer. The precore layer consists of moderately electron-dense granules of irregular shape, within moderately electron-lucent homogenous material (Fig. 14I). The granular material is localized mainly in the middle and basal region of the precore layer.

Several mature pseudocnidae situated on the apical surface of the proboscis epithelium exhibit a discharged core (Fig. 14H). The core extends parallel to the apical-basal axis of pseudocnida and is divided into two unequal parts, a short and narrow basal part and a more extended and bulbous-shaped apical part. The basal part possesses a homogenous and moderately electron dense wall surrounding a dense granular matrix (Fig. 14I), while the apical part consists of only densely packed finely granular material (Fig. 14J). The tip of the discharged core is surrounded by a mushroom-shaped cap of irregularly-shaped granules, which probably originate from granules of the precore layer. The homogenous and moderately electron-dense cortex of discharged pseudocnidae, measuring about 0.15 μm in thickness, surrounds the medulla region, and its dispersed contents consist of loosely-packed granular material (Fig. 14H).

Micrura magna

Both large and small mature pseudocnidae are present in this species, with the large type predominating (Figs. 1U, 15A and 15B). Small and large pseudocnidae are localized in different pseudocnid-forming cells. The small pseudocnidae form distinct clusters of 8–17 per cell (Fig. 15C), whereas the large pseudocnidae are observed in clusters of 5–9 (Fig. 15D).

Figure 15 Transmission electron micrographs of the pseudocnidae of Micrura magna.

(A) Panoramic view showing large pseudocnidae. (B) Longitudinal sections of small pseudocnidae. (C) Transverse section of a cluster of small pseudocnida. (D) Transverse section of a cluster of large pseudocnidae. (E) Transverse section of a large pseudocnida. (F) Longitudinal section of the apical region of a large pseudocnida. (G) Transverse sections of small pseudocnidae. co, cortex; cr, core; ico, inner cortex; me, medulla; ocl, outer core layer; oco, outer cortex; pc, precore layer; sls, spiral-like structure; sup, supportive core layer; ts, tubular structure.

Large pseudocnidae are 28.0–35.0 μm in length and up to 2.0 μm in diameter (Fig. 15A). The cortex is about 60 nm thick and is homogenous and moderately electron-dense (Fig. 15E). The medulla consists of homogenous material of somewhat greater electron-density and is about 30 nm thick near apex and middle regions and up to 1.7 μm thick in the basal region. The precore layer is homogenous and moderately electron-lucent (Figs. 15A and 15E). The long, tubular core (~20.0–25.0 μm long, up to 1.5 μm in diameter) extends from the apex to the base of the pseudocnida (Fig. 15A). The core consists of a centrally-situated, electron-dense tubular structure covered by an outer core layer composed of a moderately electron dense homogenous matrix. (Fig. 15E). At the pseudocnida apex, the outer core layer is attached to the inner surface of the cortex whereas the tubular structure merges with the cortex (Fig. 15F). The outer core layer is surrounded by an additional supportive core layer consisting of a homogenous electron-dense matrix. The outer surface of the supportive layer exhibits up to 20 helix/spiral-like structures, which extend from apex to base of the core (Fig. 15E).

The ultrastructure of both large and small pseudocnidae is similar and the two types are distinguishable only by size and the number of helix-like structures in the supportive core layer. In small pseudocnidae up to 7 are observed (Fig. 15G). The small pseudocnidae are 3.8–4.8 μm in length and 1.1–1.4 μm in diameter (Figs. 15B and 15G). Pseudocnidae with an extruded core were not observed for this species.

Notospermus albocinctus

Both large and small mature pseudocnidae are present in this species (Figs. 1V, 16A and 16B). The more abundant large pseudocnidae and rare small pseudocnidae are situated on opposite sides of proboscis epithelium. The large pseudocnidae form distinct clusters of 5–9 on the apical surface of bacillary secretory cells, whereas the small pseudocnidae are observed in clusters of 8–15 and rest on the apical surface of supportive cells.

Figure 16 Transmission electron micrographs of the pseudocnidae of Notospermus albocinctus.

(A) Panoramic view showing longitudinal sections of small pseudocnidae. The apical ends are directed upward and face away from the apical surface of the proboscis epithelium. (B) Longitudinal sections of large pseudocnidae. Orientation as in A (C) Longitudinal section of a small pseudocnida. (D) Transverse section of the small pseudocnida. (E) Longitudinal section of the apical region of a large pseudocnida. Arrowhead points to an additional thin electron dense outer layer of the cortex. (F) Transverse section of large pseudocnidae. co, cortex; cr, core; me, medulla; ocl, outer core layer; pc, precore layer; sls, spiral-like structure; ts, tubular structure.

The small pseudocnidae are ovate structures, measuring 2.0–2.3 μm in length and 1.0–1.3 μm in diameter in the basal portion of the pseudocnida (Figs. 16A and 16C). The homogenous, somewhat electron-dense cortex is about 0.2 μm thick (Figs. 16C and 16D), whereas the homogenous medulla, which exhibits greater electron density, measures about 0.25 μm in thickness. In some pseudocnidae the cortex has an additional thin electron dense outer layer, measuring about 20 nm in thickness and thus appears to be bilayered (Fig. 16E). The inner surface of the medulla is characterized by about 14 spiral-like edges that are about 30 nm thick (Fig. 16D). The precore layer is composed of a homogenous matrix exhibiting a slightly lower electron density than the cortex (Figs. 16C and 16D). The core, measuring 1.3–1.7 μm in length and 0.10–0.12 μm in diameter, extends from the apex of the pseudocnida and terminates 0.3–0.7 μm from its base (Fig. 16C). A somewhat electron-dense tubular structure is situated at the center of the core and is surrounded by an electron-dense outer core layer (Fig. 16D).

Large pseudocnidae are rod-shaped and 9.5–14 μm in length and 1.5–2.1 μm in diameter (Fig. 16B). The homogenous moderately electron-dense cortex is about 0.14 μm thick (Figs. 16E and 16F). The medulla (0.1–0.3 μm thick) is similarly homogenous but is highly electron-dense (Figs. 16B, 16E and 16F). Inner surface of the medulla possesses about 40 spiral-like edges measuring about 50 nm in thickness. The core, which is approximately 5.4–6.5 μm long and 0.7–0.8 μm in diameter, extends from the apex and terminates in the middle half of the pseudocnida (Fig. 16B). A tubular structure is situated at the center of the core and surrounded by an outer core layer (Figs. 16E and 16F). The outer surface of the core is covered by an electron-dense lamina (supportive core layer) (Fig. 16F). The tubular structure and outer core layer are both homogenous but differ in electron density: the tubular structure is moderately electron-dense, while the outer core layer exhibits a lower electron-density. Pseudocnidae with an extruded core were not observed for this species.

Hinumanemertes kikuchii

Both large and small mature pseudocnidae are present in this species (Figs. 1W, 17A and 17B). Large pseudocnidae are more abundant and are localized in an epithelial ridge along the proboscis surface (Figs. 2E and 2F), whereas small pseudocnidae are located external to the ridge. Small pseudocnidae are present in small clusters of 20–30 per cell and rest on the surface of supportive cells (Fig. 17A). The large pseudocnidae form clusters of 90–100 per cell, which rest on the apical surface of bacillary gland cell extensions (Figs. 2E and 17B).

Figure 17 Transmission electron micrographs of the pseudocnidae of Hinumanemertes kikuchii.

(A) Panoramic view of the apical surface of the proboscis epithelium showing transverse sections of a cluster of small pseudocnidae. (B) Panoramic view (as above) showing transverse sections of a cluster of large pseudocnidae. (C) Longitudinal section of a small pseudocnida. (D) Longitudinal section of the apical region of a small pseudocnida. The arrowhead indicates s an additional thin electron dense outer layer of the cortex. (E) Transverse section of middle region of a small pseudocnidae. (F) Longitudinal sections of large pseudocnidae. Arrows point to the diamond-shaped area in base of the pseudocnidae. (G) Longitudinal section of the apical region of the large pseudocnida. (H) Transverse section of apical region of a large pseudocnida. (I) Transverse section of basal region of the core of a large pseudocnida. (J) Discharged small pseudocnidae. The tip of the extruded core faces away from the apical surface of the f proboscis epithelium and is directed upward. (K) Partially discharged core of a pseudocnida. asm, apical half of submedullar region; bsm, basal half of submedullar region; co, cortex; cr, core; me, medulla; ocl, outer core layer; pc, precore layer; sup, supportive core layer; ts, tubular structure.

The small pseudocnidae are rod shaped and are 2.6–2.9 μm long and 0.9–1.0 μm in diameter (Fig. 17C). The homogenous, moderately electron-dense cortex is about 50 nm thick, whereas the highly electron-dense, homogenous medulla measures about 0.1 μm in thickness (Figs. 17C–17E). In some of the pseudocnidae of this species the cortex has an additional thin electron dense outer layer, about 25 nm thick and is thus appears bilayered (Fig. 17D). The precore layer is composed of a densely packed granular-fibrous matrix. The core is approximately 1.5–1.9 μm in length and 0.3–0.35 μm in diameter. It originates at the apex of the pseudocnidae and extends about two-thirds of its length (Fig. 17C). An electron-dense tubular structure is situated at the center of the core and is surrounded by an outer core layer with similar electron density. The tubular structure contains a thin filament, extending parallel to apical basal axis of pseudocnida (Fig. 17E). The tubular structure and the outer core layer are separated by a lucent area containing a matrix of loosely-packed material. The outer core layer is surrounded by the supportive core layer possessing highly electron-dense material (~50 nm thick) (Figs. 17D and 17E). At the apex of the pseudocnida, the outer core layer terminates on the inner surface of the cortex (Fig. 17D).

Large pseudocnidae are rod-shaped and 4.4–5.1 μm in length and 1.1–1.2 μm in diameter (Figs. 2F and 17F). The moderately electron-dense cortex is about 75 nm thick (Figs. 17G and 17H). The medulla, measuring 0.3–0.35 μm thick in the apical half and 0.12–0.14 35 μm thick in the middle half and up to 0.75 μm thick in the basal half, is similarly homogenous but is highly electron-dense (Fig. 17F). Often the basal region of the medulla contains a diamond-shaped area of moderate electron density (Fig. 17F). The submedullar region of pseudocnida is divided into two approximately equal parts (apical and basal). The apical half of submedullar region consists of a centrally situated core the basal region, of which is surrounded by a precore layer (Figs. 17F and 17G). The core, which is approximately 1.4–1.6 μm long and 0.35–0.4 μm in diameter, extends from the apex and terminates at the boundary of the apical and basal submedullar regions (Figs. 17F and 17G). The core is about one-third the length of the pseudocnida. The apical and middle core regions consist of two layers, an inner tubular structure of moderate electron density and an outer core layer of lower electron density (Fig. 17H). The basal region of the core possesses only an outer core layer (Fig. 17I). The basal half of the submedullar region consists of only the precore layer, which is homogenous and moderately electron dense (Fig. 17F). In total the precore layer encompasses approximately two-thirds of the pseudocnida length (Fig. 17F).

Some small mature pseudocnidae localized on the apical surface of the proboscis epithelium exhibit an extruded core (Fig. 17J). The discharged pseudocnida possess a moderately electron-dense wall that is about 120 nm thick surrounding a heterogeneous matrix. The inner content of the discharged core is similar in electron-density to that of the base content but appears somewhat more homogenous. The external surface of the extruded core is covered by a dense matrix (Fig. 17J). Some pseudocnidae possess a partially everted core (Fig. 17K). The wall of this core consists of the supportive core layer covered by the outer core layer.

Maculaura aquilonia

The rod-shaped pseudocnidae of M. aquilonia are 1.7–2.0 μm in length and 0.5–0.6 μm in diameter (Figs. 1X and 18A) and form small clusters containing about 12–18 per cell (Fig. 18B). The cortex measures 50–60 nm in thickness and is composed of a moderately electron-dense homogenous matrix (Figs. 18A and 18C). Some electron micrographs reveal that the cortex has an additional thin electron dense outer layer, measuring 15–25 nm in thickness, appearing thus bilayered (Fig. 18A). The medulla is comprised of a highly electron-dense homogenous substance. It is about 50 nm thick in the middle region of pseudocnida, measures up to 110 nm thick in the apical part of pseudocnida and reaches a thickness of 0.42 μm in the basal part of pseudocnida (Fig. 18A). The precore layer consists of a homogenous matrix with an electron density varying from low at the periphery to moderate internally. The peripheral precore layer reveals a thin ring of loosely packed matrix. The core is 0.9–1.1 μm long and measures about 0.2 μm in diameter, extends from the pseudocnida apex and terminates in the middle half of the organelle (Fig. 18A). The core possesses an electron-dense tubular structure surrounded by a less electron-dense outer core layer (Fig. 18C). The outer core layer and cortex merge at the apex of the pseudocnida, and both layers have a similar electron density (Fig. 18A). Pseudocnidae with an extruded core were not observed for this species.

Figure 18 Transmission electron micrographs of the pseudocnidae of Maculaura aquilonia (A–C) and Maculaura sp. (D–F).

(A) Longitudinal section of pseudocnidae. Arrow indicates loosely packed matrix in the basal part of the precore layer. The arrowhead indicates a thin electron dense outer layer of the cortex. (B) Panoramic view showing transverse section of a pseudocnida cluster. (C) Transverse sections of the pseudocnidae. (D) Panoramic view showing mature pseudocnidae on the proboscis epithelium surface. Apices of the pseudocnidae face away from the apical surface of the probscis epithelium and are oriented side upward in the figure. (E) Longitudinal sections of the pseudocnidae. Arrows point to the lamina located situated within the cortex. (F) Transverse section of a pseudocnida. Arrow points to the lamina situated within the cortex. co, cortex; cr, core; me, medulla; ocl, outer core layer; pc, precore layer; ts, tubular structure.

Maculaura sp.

Pseudocnidae are rod-shaped measuring 2.6–3.1 μm in length and about 0.7–0.9 μm in diameter (Figs. 1Y, 18D and 18E). The cortex is about 65 nm thick, uniform in composition and moderately electron-lucent (Figs. 18E and 18F). The electron-dense lamina, located inside the cortex, extends from the apical end of the pseudocnida to the basal region. The medulla (~0.1μm thick) is also homogenous and is similar in electron density to the cortex. The precore layer consists of homogenous and highly electron-dense material and it surrounds the middle region of core and extends to the basal region of the pseudocnida (Fig. 18E). It is about 70 nm thick in the region around the core and up to 0.6 μm thick in the basal region of pseudocnida. The core, measuring 1.5–1.6 μm in length and 0.3–0.4 μm in diameter, extends from the pseudocnida apex, terminating in its middle half (Fig. 18E). The core possesses a moderately electron-dense tubular structure surrounded by a less electron-dense outer core layer (Figs. 18E and 18F). The outer core layer and cortex merge at the apex of the pseudocnida, and both layers have a similar electron density. Pseudocnidae with an extruded core were not observed for this species.

Zygeupolia rubens

Pseudocnidae are rod-shaped measuring 1.7–2.3 μm in length and about 0.45–0.6 μm in diameter (Figs. 1Z and 19A) and the base or middle possesses one bulbous lateral process (Fig. 19B). Their bilaminar cortex consists of an electron-dense homogenous outer layer (about 20 nm in thick) and moderately electron-lucent homogenous inner layer (about 45 nm in thick) (Fig. 19B). The medulla, measuring approximately 0.1–0.2 μm in thickness near apex and middle regions and up to 0.4 μm thick in the basal region of pseudocnida, is composed of moderately electron-dense homogenous material (Figs. 19A–19C). The precore layer is located at the terminal end of the core and is homogenous and moderately electron-lucent (Fig. 19B). The core, measuring 1.1–1.4 μm in length and 0.35–0.45 μm in diameter, extends from the apex and terminates about 0.5 μm from the base of the pseudocnida. The core consists of an inner layer of high electron density and an outer layer of moderate electron density (Figs. 19B and 19C). The inner layer of the tubular structure is separated from the outer core layer by a moderately electron-lucent area. The outer layer extends to the apex of the pseudocnida where it merges with the cortex (Fig. 19A, inset). Pseudocnidae with an extruded core were not observed for this species.

Figure 19 Transmission electron micrographs of the pseudocnidae of Zygeupolia rubens (A–C) and Kulikovia alborostrata (D–I).

(A) Longitudinal and transverse sections of mature pseudocnidae at the apical surface of the proboscis epithelium High magnification inset shows the apex of a pseudocnida. (B) Longitudinal sections of pseudocnidae revealing a lateral process (arrowhead). (C) Transverse sections of pseudocnidae. (D) Longitudinal and transverse sections of small pseudocnidae. (E) Longitudinal sections of large pseudocnidae. (F) Transverse section of the apical region of a large pseudocnida. (G) Transverse section of the middle region of a large pseudocnida revealing a thin ring of supportive core layer (sup) encircling the core. (H) Longitudinal section of the apical region of the large pseudocnida. (I) Longitudinal sections of the middle region of the large pseudocnidae showing core and precore layer surrounded by a supportive core layer (sup). co, cortex; cr, core; ico, inner cortex layer; me, medulla; ocl, outer core layer; oco, outer cortex layer; pc, precore layer; sup, supportive core layer; ts, tubular structure.

Kulikovia alborostrata

Kulikovia alborostrata possesses both small and large rod-shaped mature pseudocnidae (Figs. 1AA, 2K, 19D and 19E). The more abundant large pseudocnidae and rare small pseudocnidae are both localized on opposite sides of proboscis epithelium. The large pseudocnidae form distinct clusters of 120–220 on the apical surface of bacillary secretory cells (Fig. 2K), whereas the small pseudocnidae are observed in smaller clusters of 25–35 and rest on the apical surface of supportive cells. The larger pseudocnidae are localized in an epithelial ridge along the proboscis surface (Fig. 2J).

The small pseudocnidae are measure 3.4–3.7 μm in length and 0.95–1.0 μm in diameter (Fig. 19D). Pseudocnidae are covered by a cortex which has a bilaminar organization: the outer layer consists of homogenous material of moderate electron-density (about 20 nm in thick), and the inner layer is composed of homogenous and moderately electron-lucent material (about 55 nm thick). The medulla is about 0.1 μm thick near apex and middle regions and up to 0.7 μm thick in the basal region and consists of homogenous, electron-dense material. The core is 1.5–1.9 μm in length, and 0.3–0.35 μm in diameter. It originates at the apex of the pseudocnidae and extends about two-thirds of its length (Fig. 19D). The core consists of centrally situated rod surrounded by an outer core layer; both exhibit a similar density.

The large pseudocnidae measure 4.0–4.7 μm in length and 0.55–0.65 μm in diameter (Figs. 2J and 19E). The moderately electron-lucent homogenous cortex is 45–55 nm in thick (Figs. 19F–19H). In some pseudocnidae an additional thin electron dense outer layer, measuring 15–20 nm in thickness is present, imparting a bilayered appearance to the cortex (Fig. 19F). The medulla is up to 96 nm thick near the apex of pseudocnida and up to 0.47 μm in the basal region of pseudocnida (Fig. 19E). It consists of a moderately electron-dense homogenous matrix. The precore layer is located at the terminal end of the core and is homogenous and electron-dense (Figs. 19E and 19I). The core measures 1.6–1.8 μm in length and 0.30–0.35 μm in diameter and extends from the apex to the middle half of the pseudocnida (Figs. 19E and 19H). It consists of homogenous layers with different electron density. The inner layer possesses a tubular structure with moderate electron density, and the outer core layer is moderately electron-lucent. The middle and basal halves of the outer core layer and precore layer are surrounded by a thin tubular structure consisting of dense material (supportive core layer; Figs. 19G and 19I).

Pseudocnidae with an extruded core were not observed for this species.

Cerebratulus cf. marginatus

The epithelial surface exhibits numerous clusters of pseudocnidae; each cluster contains 220–300 pseudocnidae (Fig. 2I). Pseudocnidae are rod-shaped, measuring 3.1–4.3 μm in length and up to 1.0 μm in diameter (Figs. 1BB and 20A). The cortex measures about 80 nm in thickness and consists of moderately electron-lucent homogenous material (Fig. 20B). The medulla is 0.1–0.2 μm thick in the apical and middle part of pseudocnida and up to 0.8 μm thick in the basal part of pseudocnida (Figs. 20A and 20C). It consists of a moderately electron-dense homogenous matrix. The precore matrix layer exhibits regions of both high and low electron density (lucent; Fig. 20A). The lucent region occupies a broad area around the terminal end of core. The core extends from the apex and terminates in the middle part of pseudocnida (Fig. 20A). The core possesses a moderately electron-dense tubular structure surrounded by a less electron-dense outer core layer (Figs. 20B and 20C). The outer layer extends to the apical surface of the pseudocnida where it merges with the cortex (Fig. 20C). Pseudocnidae with an extruded core were not observed for this species.

Figure 20 Transmission electron micrographs of the pseudocnidae of Cerebratulus cf. marginatus (A–C), Lineus sanguineus (D and E) and Lineus viridis (F–I).

(A) Longitudinal section of a pseudocnida. The precore layer forms a dense “cup-like” structure, inside of which the lucent region occupies a broad area around the terminal end of the core (asterisk). (B) Transverse section of a pseudocnida. (C) Longitudinal section of the apical region of a pseudocnida. (D) Longitudinal section of pseudocnidae. Arrowhead indicates the thin electron dense outer layer of the cortex. (E) Transverse section of a pseudocnida. (F) Panoramic view showing mature pseudocnidae released on the proboscis epithelium surface. Apices of the pseudicnidae are oriented towards the top of the figure. Their bases rest on underlying gland cells (asterisks). (G) Longitudinal section of a pseudocnida. (H) Transverse section of the apical region of the pseudocnida. (I) Transverse section of the middle region of a pseudocnida. co, cortex; cr, core; me, medulla; ocl, outer core layer; pc, precore layer; ts, tubular structure.

Lineus sanguineus

Pseudocnidae are rod-shaped and measure 2.6–2.8 μm in length and about 0.5–0.6 μm in diameter (Figs. 1CC and 20D). The cortex is about 45 nm thick and is homogenous and moderately electron-lucent (Fig. 20D and 20E). In some micrographs an additional thin electron dense outer cortex layer, about 15–20 nm thick can be discerned (Fig. 20D) and it gives the cortex is a bilayered appearance. The medulla (~0.1–0.17 μm thick) is also homogenous but is slightly more electron-dense than the cortex. The precore layer is located at the middle and terminal end of the core and consists of a homogenous, electron-dense material (Fig. 20D). Its thickness varies from about 90 nm thick in the middle region to 0.35 μm thick in the basal region of pseudocnida. The core, measuring 1.1–1.3 μm long and 0.3–0.4 μm in diameter, extends from the pseudocnida apex towards its base, terminating in the middle half of the pseudocnida (Fig. 20D). The core possesses a moderately electron-dense tubular structure surrounded by a less electron-dense outer core layer. The outer core layer and cortex merge at the apex of the pseudocnida, and both layers have similar electron density (Fig. 20D). Pseudocnidae with an extruded core were not observed for this species.

Lineus viridis

The pseudocnidae are rod-shaped and are 2.5–4.1 μm in length and 0.8–0.9 μm in diameter (Figs. 1DD and 20F). The cortex is about 50 nm thick and is composed of a moderately electron-lucent homogenous substance (Figs. 20G and 20H). In some of the pseudocnidae of this species the cortex has additional thin electron dense outer layer, about 20 nm thick making it appear bilayered (Fig. 20G). The medulla measures approximately 0.1 μm in thickness in the apical and middle half of pseudocnida and up to 0.4 μm thick at its base. This region is homogenous and exhibits moderate electron-density (Fig. 20G). The precore layer consists of homogenous and highly electron-dense material and surrounds the middle and base of the core (Figs. 20G–20I). The core, measuring 0.9–1.1 μm in length and about 0.25 μm in diameter, extends from the apex of the pseudocnida and terminates in its midregion (Fig. 20G). A tubular structure is situated at the center of the core and is surrounded by an electron-lucent outer layer (Fig. 20H). The cortex and outer core layer exhibit a similar electron density. Pseudocnidae with an extruded core were not observed for this species.

Discussion

Size, shape and location of pseudocnidae

The pseudocnidae are usually rod- or club-shaped structures, but some Hubrechtella species, including H. ijimai, have unusual spherical-shaped pseudocnidae, which previously were called “spherical bodies” (Kajihara, 2006; Chernyshev et al., 2017). In most species, pseudocnida size varies from 1.5 to 5.5 μm in length and from 0.7 to 1.7 μm in diameter. The smallest pseudocnidae (0.6–0.7 μm in length and 0.4–0.5 μm in diameter at their base) were revealed in Tubulanus cf. pellucidus, and the longest pseudocnidae—in Micrura magna (25 μm long) and Notospermus albocinctus (35 μm long). Similarly long rod-shaped pseudocnidae were previously observed in Notospermus geniculatus (up to 30 μm length, see (Riser, 1991)), and in heteronemertean Antarctolineus scotti (up to 100 μm or more in length, see (Gibson, 1985)). Long pseudocnidae were also described for H. juliae (14 μm, see (Chernyshev, Magarlamov & Turbeville, 2013)) and H. queenslandica (33 μm, see (Gibson, 1979)).

In the majority of the species investigated, only one type of pseudocnida is present, but there are a number of palaeo- and heteronemertean species with two pseudocnida types (dimorphic pseudocnidae) that are distinguished primarily by size (large vs. small) but also in some species by variation in substructure. Pseudocnida dimorphism occurs in C. mutabilis, H. kikuchii, K. alborostrata, M. magna, and N. albocinctus; in the latter two species, the differences in size of each type was the most significant. Pseudocnida dimorphism was previously revealed as a common feature of Cephalothrix species (Magarlamov, Chernyshev & Turbeville, 2018).

The pseudocnidae of most nemertean species examined to date are organized in clusters containing from 20 to 300 and more densely packed. Each cluster is formed by a single pseudocnida-forming secretory cell that consists of a broad cell body and a narrow neck, the distal portion of which forms a cushion-shaped papilla resting on the epithelial surface (Chernyshev, Magarlamov & Turbeville, 2013; Montalvo et al., 1998; Turbeville, 2006; current research). The biggest pseudocnida clusters observed are present in Parahubrechtia sp. (>300 pseudocnidae per cluster; Fig. 7B), Cerebratulus cf. marginatus (220–300 pseudocnidae per cluster; Fig. 2H) and K. alborostrata (up to 220 pseudocnidae per cluster; Fig. 2I). In species with dimorphic pseudocnidae the large and small forms occur in clusters of differing size. In M. magna, small pseudocnidae occur in clusters containing 8–17 per cell (Fig. 15C), whereas large pseudocnidae are grouped in clusters of 5–9 per cell (Fig. 15D). For species of the genus Cephalothrix, large pseudocnidae are found in clusters of 2–6 and rarely 10–20 or more, while small pseudocnida clusters contain 20 per cell (Magarlamov, Chernyshev & Turbeville, 2018). Pseudocnida of clusters in cephalotrichids are not packed as densely as in other nemerteans, and may exhibit an oblique arrangement, particularly when most of their length extends above the cell apex (Fig. 2B). The pseudocnidae of H. ijimai (current research) and H. juliae (see Fig. 2G in Chernyshev, Magarlamov & Turbeville (2013)) are scattered or arranged in small groups of 2 or 4. The presence of small groups containing 4–5 pseudocnidae has also been shown for Riserius pugetensis (Norenburg, 1993)

Comparative analysis of pseudocnida ultrastructure

The outer layer (cortex) of pseudocnidae covers all but the apex of the structure. The cortex consists of two sublayers of different electron density in C. mutabilis (large pseudocnidae), Heteronemertea sp.5DS, H. ijimai, H. juliae, M. ignea, N. uchidai, Nipponomicrura sp., M. kulikovae, and M. cf. bella. In contrast, the cortex of small pseudocnidae of C. mutabilis, the outer cortex layer of large pseudocnidae of C. mutabilis and the cortex of Maculaura sp. have a thin and dense layer dividing it into two sublayers equal in density. In some species examined, some micrographs show a cortex consisting of a dense, thin outer layer and a less dense and thicker inner layer. However, this is likely attributable to fixation artifacts. In other species, the cortex is single-layered. In most species, the cortex is 30–120 nm in thickness, but in large pseudocnidae of N. albocinctus, the thickness of this layer reaches 200 nm, and in H. ijimai and H. juliae—up to 300 and 450 nm respectively. Situated directly internal to the cortex is the subcortical layer or medulla. In the majority of species, this layer occupies the largest volume of the pseudocnida. Exceptions include the medulla of Balionemertes australiensis, Cephalotrichella echinicola, M. magna, small pseudocnidae of Hinumanemertes kikuchii and K. alborostrata (current research), and large pseudocnidae of Cephalothrix cf. rufifrons (Magarlamov, Chernyshev & Turbeville, 2018). In these species the medulla forms a thin layer around an expanded precore layer. In Tubulanus sp. IZ-45559, Parahubrechtia sp., Tubulanidae sp. Kurambio2-17, the medulla consists of concentric electron-dense membranes, separated from each other by an electron-transparent material. In N. albocinctus, the inner surface of the medulla is characterized by a helix-like structure that may serve a supporting function. A similar structure was found in the pseudocnidae of M. magna, however, in this species this structure surrounds the core.

Situated internal to the medulla, is the precore layer, which is the most variable layer of pseudocnidae. Even in closely related species it can vary in electron density, and volume (see the description of small pseudocnidae in C. simula and C. hongkongiensis in Magarlamov, Chernyshev & Turbeville (2018)). In some species, the precore layer is poorly defined (T. cf. pellucidus, Tubulanus sp. IZ-4559, Parachubrechtia sp., Tubulanidae sp. 33DS) or absent (T. punctatus, N. uchidai). The precore layer of Carinomella lactea, M. ignea, M. kulikovae, M. cf. bella, and large pseudocnidae of H. kikuchii occupies the middle to basal part of the pseudocnidae and only its apical end contacts the core. In M. ignea, the contact point of the precore layer with the core is reinforced by an additional ring of electron-dense material.

In most species investigated, the precore layer is relatively wide and surrounds the core. In M. aquilonia, Maculaura sp., Cerebratulus cf. marginatus, L. sanguineus, L. ruber, L. viridis, K. alborostrata (large pseudocnidae), and R. occultus the precore layer consists mainly of highly electron-dense material and forms the so-called “electron-dense socket/pocket” (Ling, 1971) or “cup-shaped layer” (Montalvo et al., 1998). In the remaining species, the precore layer consists of a homogeneous material with medium or low electron-density. In five studied species (Tubulanidae gen. Sp. IZ-45557, Tubulanidae sp. Kurambio2-17, Heteronemertea gen. sp.5DS, M. kulikovae, M. cf. bella), the homogeneous material of the precore layer also contains electron-dense granules. In B. australiensis and C. echinicola (Magarlamov, Chernyshev & Turbeville, 2018), the precore layer is divided into two sublayers, an apical precore layer and a basal layer. The former layer surrounds only the core, whereas the latter layer fills the basal part of the pseudocnida. The two sublayers also differ in electron density.

The innermost layer of pseudocnidae is referred to as the core and it occupies a central position, except in H. juliae, where it is shifted to the periphery (Chernyshev, Magarlamov & Turbeville, 2013). In all species examined, the core is straight in appearance. Although it appears as a coiled filament in light micrographs of pseudocnidae of some Cephalothrix species (see figures in (Gerner, 1969)), in our opinion, this configuration may result from fixation artifact. The core structure, as a rule, is similar in closely related species, but significantly differs in phylogenetically distant species. Based on the structure of the core, we distinguished two pseudocnida types: type I pseudocnida are characterized by a core in the form of a tube-like structure surrounded by an outer core layer (i.e., a two-layer core); type II pseudocnida possess a core consisting of a centrally located rod or a tube-like structure, surrounded by inner (core rod lamina) and outer core layers (i.e., a three-layer core). Type I pseudocnidae were found in archinemertean species (Magarlamov, Chernyshev & Turbeville, 2018; Turbeville, 2006, current research), C. mutabilis, and in almost all Pilidiophora, except Heteronemertea sp. 5DS. However, in the small pseudocnidae of C. mutabilis, the tube-like layer is absent (Fig. 3D). Pseudocnidae of two species of Nipponomicrura, M. magna, Z. rubens, the small pseudocnidae of H. kikuchii, and large pseudocnidae of C. mutabilis have an additional layer represented by high electron-dense material surrounding the core referred to as the supportive core layer. In small pseudocnidae of H. kikuchii, this layer takes part in eversion and strengthening of the extruded core. This may also apply to other pseudocnidae with a supportive core layer. In small pseudocnidae of C. filiformis sensu Iwata, the core is rudimentary relative to those of other nemertean pseuodcnidae and is a rod-shaped structure (Magarlamov, Chernyshev & Turbeville, 2018). Type II pseudocnidae were found in all studied species of Tubulanidae and the basal Heteronemertea sp. 5DS.

Pseudocnida extrusion mechanism

Extruded (= discharged) pseudocnidae were previously detected in some species (Martin, 1914; Gerner, 1969) and we observed them in eleven species of palaeonemerteans and in eight species of pilidiophorans (Chernyshev, Magarlamov & Turbeville, 2013; Magarlamov, Chernyshev & Turbeville, 2018; current research). In all the deep-sea species (Tubulanidae sp. IZ 45557, Tubulanidae gen. sp. 33DS, Tubulanidae sp. Kurambio2-17, and Heteronemertea sp. 5DS), most of the mature pseudocnidae were discharged, which likely can be explained by the pressure drop during material collection. The structure of pseudocnidae undergoes significant changes during the extrusion process. The inner content of a discharged pseudocnida becomes homogeneous and consists of either a loose or tightly packed fibrillar matrix in archinemertean species (Magarlamov, Chernyshev & Turbeville, 2018), Tubulanidae sp. IZ 45557, Tubulanidae gen. sp. 33DS, Tubulanidae sp. Kurambio2-17, Heteronemertea sp. 5DS, and H. juliae or a medium or high-density matrix as in T. punctatus, Parahubrechtia sp., N. uchidai, and M. cf. bella. In contrast the inner content of discharged pseudocnidae consists of densely packed, highly electron-dense granular material enclosed in a homogeneous material of medium electron density in C. mutabilis.

Based on the structure of the core in discharged pseudocnidae in 10 of 32 species investigated, we distinguish three different mechanisms of core extrusion. Mechanism (1) occurs in type I pseudocnidae of most species, mechanism (2) typifies type I pseudocnidae of M. cf. bella, and the third mechanism (3) characterizes type II pseudocnidae. In type I pseudocnidae, the extrusion mechanism is associated with the outer core layer (Fig. 21A). In undischarged pseudocnidae, the outer core layer and cortex in the apical part of the pseudocnidae are confluent, and both of these layers exhibit a similar density. During core extrusion, the outer core layer everts and, simultaneously, the tube-like layer becomes the outer layer (Fig. 21A). In the small pseudocnidae of H. kikuchii, the supportive core layer everts along with the outer core layer (Fig. 17K); in this case, in the discharged pseudocnidae, the supportive core layer is situated internally, and the outer core layer externally. During the extrusion of type I pseudocnidae, the diameter and length of the extruded core increases by 1.25–1.5 and 2–2.5 times, respectively, compared to the core in non-extruded pseudocnidae. The increase in diameter is associated with a tube-like core, which becomes externalized and covers the wall of the extruded core at the end of the extrusion process. An increase in the length of the extruded core associated with its extension, occurs in the pseudocnidae of C. mutabilis, H. juliae, N. uchidai, Nipponomicrura sp., and H. kikuchii (small pseudocnidae). The fact that the extruded core of heteronemerteans forms a very long “thread” was noted earlier (Martin, 1914; Norenburg, 1993). The inner content of the discharged pseudocnida core varies. It may be filled with wrinkled membranous material (C. echinicola (Magarlamov, Chernyshev & Turbeville, 2018)), dense granular material (C. mutabilis (current research), C. simula and C. filiformis sensu Iwata (Magarlamov, Chernyshev & Turbeville, 2018)) or fine homogenous material (H. juliae, Nipponomicrura sp., N. uchidai, M. cf. bella and H. kikuchii (current research)).

Figure 21 Schematic diagrams of pseudocnidae core extrusion mechanisms.

(A) Mechanism I illustrated for type I pseudocnidae with the small pseudocnidae of Carinoma mutabilis. (B) Mechanism II illustrated with pseudocnida of Micrura cf. bella. (C) Mechanism III in type II pseudocnidae illustrated with pseudocnida of Tubulanus punctatus).

In Cephalothrix species, core extrusion is also associated with the eversion of the outer core layer as described above for type I pseudocnidae. In the three Cephalothrix species (C. atlantica, C. pacifica, and C. mediterranea), the extruded core extends 2–4 times the length of undischarged pseudocnida (Gerner, 1969). In C. germanica (see Abb. 10B in Gerner (1969)), C. simula, C. filiformis sensu Iwata, and C. cf. rufifrons (see Figs. 6G, 7C, and 8E in (Magarlamov, Chernyshev & Turbeville, 2018)), the extruded core is comparable in length to undischarged pseudocnidae.

In M. cf. bella, the extrusion mechanism differs from that observed for other species with type I pseudocnidae (Fig. 21B). The extruded core consists of a short, narrow distal region and a more elongated and bulb-shaped proximal region. The distal region of the extruded core is formed by the outer core layer, and the proximal region—by the medullar layer, the contents of which pass through the extruded core, forming a club-shaped or oval structure at its apex. The material of the precore layer forms an amorphous material around the top of the extruded core. It is plausible that this extrusion mechanism also occurs for large pseudocnidae of M. ignea, M. kulikovae, and H. kikuchii, given that they possess a structurally similar core structure and submedullar region.

In type II pseudocnidae, the process of core extrusion begins with the eversion of the outer core layer together with the core rod and associated core rod lamina from the inner space of the pseudocnidae (Figs. 5F and 21C). Thus, after the complete eversion of the outer core layer, the inner cavity of the extruded core is filled with a core rod wrapped with the core rod lamina (T. punctatus) or with the loose fibrillar material (Heteronemertea sp. 5DS). The inner content of the extruded core is partially released to the outside, and probably serves to increase the surface area for possible adhesion to the prey.

Different mechanisms of core extrusion suggest functional differences between type I and II pseudocnidae, but we can only judge this indirectly. There are no documented observations of pseudocnidae discharge during the prey capture. Although Jennings and Gibson (Jennings & Gibson, 1969) report that pseudocnidae (barbs) of Cephalothrix bioculata and C. linearis penetrate the prey’s integument, it was not confirmed by their light microscopic images. While it may be possible that the pseudocnidae of Cephalothrix and those of Cephalotrichella and Balionemertes are able to penetrate the prey’s integument, we did not observe any specific structures that support a penetration role.

An adhesive function for pseudocnidae suggested by various authors (Jennings & Gibson, 1969; Montalvo et al., 1998; Turbeville, 2006) seems more realistic to us, although no direct evidence of this has yet been obtained. The presence of the fibrillar-granular material on the outer surface of the extruded core in Cephalothrix and Cephalotrichella species (Magarlamov, Chernyshev & Turbeville, 2018), as well as in Tubulanidae sp. IZ 45557, Tubulanidae sp. Kurambio2-17, Parahubrechtia sp, Heteronemertea sp. 5DS, Nipponomicrura uchidai, and small pseudocnidae of Hinumanemertes kikuchii suggests that these structures may have an adhesive function. The pseudocnidae of T. punctatus and M. cf. bella have a mushroom-shaped extruded core that most likely functions in adhesion. Although an adhesive function seems reasonable based on structural evidence, it is noteworthy that L. sanguineus is capable of rapidly immobilizing its prey with a brief proboscis strike that does not appear to involve obvious adhesion, suggesting alternatively that the pseudocnidae of this species may release a toxin that diffuses across the body wall of the annelid prey (Caplins & Turbeville, 2011). Except for the detection by polyclonal antibodies to TTX-like substances in Kulikovia alborostrata (= Lineus alborostratus; (Magarlamov, Shokur & Chernyshev, 2016)) data on potential pseudocnidae toxins content is absent. Clearly, more research is needed to elucidate precisely how these organelles are utilized.

Pseudocnidae maturation

Even though pseudocnidae cannot be reused after core extrusion and thus, require replacement, we identified maturing pseudocnidae only in H. juliae (Chernyshev, Magarlamov & Turbeville, 2013), C. simula (Magarlamov, Chernyshev & Turbeville, 2018), B. australiensis, and Heteronemertea sp. 5DS (current research). Previously, the initial stages of pseudocnidae maturation were described for L. ruber (Ling, 1971), and Riseriellus occultus (Montalvo et al., 1998). The process of pseudocnida maturation begins with the formation of a rather large secretory granule with homogeneous (R. occultus) or heterogeneous content (other species). The secretory body forms in the expanded cisternae of the rough endoplasmic reticulum (RER); however, vesicles of the Golgi apparatus can also contribute to its formation (see (Ling, 1971)) (Fig. 22A). In subsequent maturation stages, the secretion products of the immature pseudocnidae segregate into separate layers: a rather thin envelope, which subsequently separates into the cortex and medulla, and the extensive internal layer (Figs. 22B and 22C). At this stage, in B. australiensis lateral processes of expanded RER cisternae were detected (Figs. 4D and 22B). Ling (Ling, 1971) also noted the presence of a well-developed network of the endoplasmic reticulum (ER) around immature pseudocnidae of L. ruber. In mature pseudocnidae, lateral processes remain, but their contents are represented by a homogeneous medium or high-density product. We speculate that lateral processes participate in the growth of a pseudocnida and diminish in size as in most species after their formation. Their persistence was interpreted as a synapomorphy of Archinemertea (Magarlamov, Chernyshev & Turbeville, 2018). The core forms as a small process extending inward from the apex of the pseudocnidae (Figs. 22B and 22C). Data on the formation of pseudocnidae reveal the origin of their layers and support the identification of four layers adopted in this work.

Figure 22 Locality, number of specimens of the nemerteans examined.

(A) The formation of a secretory granule from the expanded cisternae of the rough endoplasmic reticulum (RER) and vesicles of the Golgi apparatus (GA). (B) Stages of pseudocnidae maturation in Balionemertes australiensis. (C) Stages of pseudocnidae maturation in Heteronemertea gen. sp. 5DS. ER, endoplasmic reticulum; RER, rough endoplasmic reticulum; GA, Golgi apparatus.

Pseudocnidae and other metazoan extrusomes

The basic structure of mature cnidae (e.g., nematocysts, spirocysts) of Cnidaria and the pseudocnidae of Nemertea are convergently similar, consisting of a body with a long inverted tubule attached at the apical end. However, the core of the pseudocnida is never folded or rolled into a spiral state being lengthened at discharge only by elongation of the outer core layer. The inverted tubule of cnidae (Reft & Daly, 2012) and the core of pseudocnidae are attached to the capsule wall in a similar way. The length of cnidae varies from 5 to 100 μm (Mariscal, 1974), which is comparable in size to large pseudocnidae, although in most species investigated, pseudocnidae are smaller (0.55–4.8 μm long). The nematocyst capsule wall and tubules contain minicollagens and other proteins (Beckmann et al., 2015; Bentele et al., 2019), but the structural compounds of pseudcnidae have not been characterized. The ultrastructural data show that the capsule, as well as the cortex of pseudocnidae of some nemertean species, consist of narrow outer and wide inner layers. The obvious differences are in the structure of the inner layer. In pseudocnidae this layer consists of homogeneous, moderately electron-dense or electron-lucent material, whereas in the wall of nematocysts, it consists of electron-lucent material traversed by fine fibrils (Westfall, 1966; Campbell, 1977). The cortex of the pseudocnidae of most nemerteans consists of a single wall. The envelope of the cnidarian spirocysts also has a single-wall, but it is a thin with closely-spaced ridges or serrations on the inner surface, the tips of which project into the lumen (Reft & Daly, 2012; Westfall, 1966; Mariscal, 1974). In undischarged cnidae, the tubule wall consists of two layers, a (thin electron-dense outer layer and a thicker and moderately electron-dense inner layer (Tardent, 1988). In contrast, the core of pseudocnidae consists either of a centrally located tube-like structure or rod covered by an outer core layer. Cnidae contain apical structures (operculum, apical flaps, or apical cap) that seal the opening of the tube and capsule (Reft & Daly, 2012; Östman, 1982; Reft, Westfall & Fautin, 2009), and under appropriate stimulation, the apex of the capsule opens and the tubule everts from the capsule (Reft & Daly, 2012; Mariscal, 1974). Also, microtubules encircle the capsule apex in some cnidae (Watson & Mariscal, 1985). The pseudocnida tip/apex lacks any structures overlying the core opening and no associated microtubules are observed. The triggering mechanism of cnidarian nematocysts is supported by cilium/cnidocil that is a mechano- and chemo-receptor located on the apical surface of the cell. pseudocnida-bearing cells do not possess any ciliary structures on their surface Pseudocnida discharge is likely triggered by mechanical stimulation.

The pseudocnidae of many nemerteans exhibit basic similarities with nematocysts that perform a penetration function (penetrants). Both structures have a uniform diameter, and straight discharge path of their eversible structures and the base of these remains attached to the wall of the capsule post eversion (Colin & Costello, 2007). In nemerteans, the length of the everted core of pseudocnidae varies from 1 μm (Carinoma mutabilis) to 40 μm (Hubrechtella juliae see Fig. 2b in Chernyshev, Magarlamov & Turbeville (2013)), and in nematocysts-penetrants the thread is long, and for example, varies from 115 μm in Metridium senile (Östman et al., 2010) to 470 μm or more in Hydra attenuata (Tardent et al., 1985). The everted nematocysts thread can be armed with spines and divided into a dilated portion or proximal shaft and a thread-like distal tubule (Reft & Daly, 2012; Colin & Costello, 2007; Fautin & Mariscal, 1991; Östman, 2000), and be capable of penetrating even the hard cuticle of crustaceans. In contrast the core surface of nemertean pseudocnidae is smooth and undifferentiated along its length, indicating that, most likely, it does not penetrate nor anchor to the prey integuments.

Cnidae and pseudocnidae maturation and subsequent growth share some common general features, but details differ. Both secretory bodies (Westfall, 1966; Beckmann & Özbek, 2012; Okamura, Gruhl & Reft, 2015) are formed by either ER or the Golgi apparatus, and further growth is attributed to fusion of smaller vesicles. The fusion of Golgi apparatus/ER vesicles of the growing cnida vesicle and immature pseudocnida occur at a specific region. In cnidae, it is at the apical tip of the growing tubule (Slautterback, 1961; Adamczyk et al., 2010; Özbek, 2011), and in pseudocnidae it is the lateral process (current research). Although an everted tubule of a cnidae and the everted core of a pseudocnida are situated at the apical end of the granule, subsequent tubule growth is markedly different. In immature cnidae the tubule forms as an outgrowth of the forming vesicle, and it is stabilized by microtubules (Adamczyk et al., 2010). The extended tubule subsequently invaginates and is then tightly packed inside the capsule body (Westfall, 1966; Östman et al., 2010; Adamczyk et al., 2010). In pseudocnidae, the forming tubule grows inside the body of the granule and no microtubules are involved. In summary comparison of the morphology and development of cnidae and pseudocnidae reveal only superficial similarities, strongly supporting these as convergently similar structures and thus with additional data, confirming earlier studies (Martin, 1914; Turbeville, 2006).

In addition to cnidae, a variety of epidermal extrusomes are known among acoelomorph turbellarians and rhabditophoran flatworms, but none of these exhibits strong resemblance to pseudocnidae. The most studied are rhabdites, which are readily distinguished from pseudocnidae by a striated lamellate cortex and the absence of a central eversible structure (filament/core) (Smith et al., 1982). Although differing in shape, sagittocysts of some Acoela resemble pseudocnidae in substructure, consisting of three layers: an electron-lucent core filament (’needle’) with rhomboidal or circular cross-sectional profile, an intermediate layer of electron-lucent material and a more electron-dense cortex of finely fibrous material (Yamasu, 1991; Gschwentner et al., 1999). In contrast to pseudocnidae an associated muscle (muscle mantle) surrounds much of the sagittocysts with the exception of the small-type found associated with male copulatory organs in one species (Yamasu, 1991). The mechanism of sagittocyst extrusion also differs from that of pseudocnidae: contraction of the special muscle mantle around terminal part of the sagittocyte leads to rapid expulsion of the sagittocyst (Gschwentner, Baric & Reinhard, 2002). The so-called paracnids of Proseriata are cellular structures with a centrally located eversible tube, but this is part of the modified secretory cell rather than a secretory granule (Sopott-Ehlers, 1981). Other paracnids consist of a secretory cell containing membrane-bound granules without a tubular core or filament (Sopott-Ehlers, 1985). Also, various rhabdoids of plathelminths (Smith et al., 1982; Rieger et al., 1991) exhibit morphologies unlike those of nemertean pseudocnidae.

Implications for systematics

Our comparative analysis revealed morphological diversity in nemertean pseudocnidae, although, as a rule, the pseudocnidae of closely related species have high ultrastructural similarity. The nemertean proboscis contains one or two types of pseudocnidae, the extruded cores of which possess a smooth external surface lacking spines or other extensions. Because of the somewhat limited morphological variation, pseudocnidae do not offer as great a source of informative features for systematics as the analogous cnidae of Cnidaria (Fautin, 2009). However, TEM studies revealed significant differences in the arrangement of layers in pseudocnidae, and we showed that some features of pseudocnida ultrastructure can provide a useful source of additional characters for nemertean systematics. For example, some of these novel features distinguish Cephalotrichellidae and Cephalotrichidae (Magarlamov, Chernyshev & Turbeville, 2018). The data obtained for Riseriellus occultus, Lineus viridis, and Lineus sanguineus reveal that their pseudocnidae are very similar (current research), but these nemerteans have different feeding biologies (McDermott & Roe, 1985; Caplins & Turbeville, 2011), suggesting evolutionary conservation of pseudocnidae in phylogenetically closely related species. Ultrastructure of the pseudocnidae of heteronemertean species is of special interest since traditional internal morphological characteristics of the body and proboscis do not reveal reliable synapomorphies for phylogenetic analyses. The data strongly support homology of pseudocnidae among nemerteans consistent with previous studies. Regarding their distribution in the nemertean phylogeny (Kvist, Chernyshev & Giribet, 2015; Andrade et al., 2014), a plausible interpretation is that this character was present in their common ancestor and later lost in several lineages. Their distribution also suggests independent evolution of pseudocnida dimorphism in the Palaeonemertea and Pilidiophora. A detailed evaluation of pseudocnida evolution within Nemertea is in preparation.

We thank Neonila E. Polyakova (National Scientific Center of Marine Biology, Far Eastern Branch, Russian Academy of Sciences) and Svetlana Maslakova (Oregon Institute of Marine Biology) for supplying several specimens of some nemertean species.

Additional Information and Declarations

Competing Interests

Author Contributions

Data Availability

The authors declare that they have no competing interests.

Timur Yu Magarlamov analyzed the data, prepared figures and/or tables, and approved the final draft.

James M. Turbeville performed the experiments, authored or reviewed drafts of the paper, and approved the final draft.

Alexei V. Chernyshev conceived and designed the experiments, authored or reviewed drafts of the paper, and approved the final draft.

The following information was supplied regarding data availability:

Raw data are available at Figshare:

Magarlamov, Timur; Turbeville, James; Chernyshev, Alexei (2020): raw.zip. figshare. Figure. DOI 10.6084/m9.figshare.12899840.v1.

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
