# Peer review of "Pseudocnidae of ribbon worms (Nemertea): ultrastructure, maturation, and functional morphology"

_PeerJ, doi:10.7717/peerj.10912_

## Round 0.1 · original submission · Minor Revisions

Dear authors,

Many thanks for your submission to PeerJ and I apologize for the delay in getting the reviews back to you. As you may find in the attached reviews, both reviewers were very please with the content of your manuscript and have provided many suggestions to improve it. One of the reviewers has also indicated that some parts of the manuscript need additional language review and I would like you to pay special attention to that.

·

Basic reporting

This ms seeks to present a comparative analysis, based primarily on transmission electron microscopy, of the enigmatic pseudocnidae found in the proboscis of certain nemertean worms. It presents a great deal of new data as well as reinterpretation of earlier observations and succeeds admirably at presenting what is likely to be the definitive understanding of these structures. Organization of the ms is clear and language structure is excellent.

Experimental design

Experimental design is thorough, encompassing a suitable diversity of taxa for the intended comparative analysis.

Validity of the findings

The findings are convincing and fully justified by text and figures.

Additional comments

My only misgiving is a lack of reference to the unusual proboscis of Heteroenopleus enigmaticus Wern, 1998. Perhaps the authors cannot infer anything about the "stylet" structures described but I think it would be worth stating that and why.
In the attached document, I have highlighted words where there is a typographical error, a missing comma, or something else. There are many missing "the" (an understandable problem of native Russian speakers writing in English), most of which I think I marked but I doubt that I caught all of them.

Reviewer 2 ·

Basic reporting

The introduction shows the background of the study and comprehensively covers the relevant published information. Generally, the overall structure of the manuscript conforms to the standards of the journal. Some details, with respect to phrasing and labelling of images, however need to be addressed as problematic:
Text: The text, especially in the, very extensive, results-section shows several inconsistencies with respect to phrasing and grammar. I marked some of these inconsistencies, but as I am not a professional language editor some of the phrasing inconsistencies might have slipped my attention as well. I attribute that to the corresponding author not being a native speaker and would strongly recommend the manuscript being language edited, at least by the second author, who is a native speaker.
Figures: The figures are largely well and sufficiently labeled. However, the usage of vertical scale bars is absolutely unusual and severely disturbs the viewing experience. I would strongly prefer this to be changed to horizontal scale bars throughout. Some images (commented in the commented pdf-file of the manuscript) in nearly all figure plates are under-labeled, thus leaving the reader lost in the respective image – these images need sufficient labeling. Additionally, some figure captions are incomplete or redundant (marked throughout the commented pdf-file of the manuscript as well). In the text, the images are generally referred to, in some cases additional reference to images should be made for clarification (see comments in commented pdf-file of manuscript). Raw data has not been supplied by the authors but the supplied images are fully sufficient to illustrate the described findings.

Experimental design

The presented data are original and within the scope of the journal. The research question is defined in that the aim is to provide a comprehensive account on the comparative ultrastructure of nemertean pseudocnidae. As this aim is purely descriptive, a research question in the strict sense cannot be defined. Nevertheless, the data presented are relevant and meaningful as they provide a sound baseline knowledge for future research addressing the evolution and ecological significance of the described structures that have been known to represent an important but still poorly known aspect of nemertean anatomy.
Technically, the data acquisition is done on an advanced level as all three authors are known to be experienced ultrastructure-researchers. Unfortunately, some images have folds and not all show optimal contrast for a reader unexperienced in ultrastructural research to assess all details described in the text. The methods employed are fairly standard methods and are largely described to enough detail to be replicated (requests to specify a few aspects have been included by me in a commented pdf-file of the manuscript).

Validity of the findings

The presented data is the most comprehensive account on the ultrastructure of pseudocnidae in Nemertea ever published and thus provides a very valuable contribution to biological science. The sub-sections of the discussion seem a bit unbalanced in my opinion, since there is a comparably long discussion on the similarities and differences of cnidarian cnidae and nemertean pseudocnidae, just to make the point that these are not homologous structures. On the other hand, the extrusomes of more closely related taxa are only briefly touched. The manuscript would benefit from a more balanced account. Conclusions on the systematic implications on the other hand are rather brief and do not take all results into account at equal rate (e.g. the lateral process in the heteronemertean species Z. ruber). Comments on that issue can be found in the commented pdf-file of the manuscript. Clearly, the largest profit of this manuscript is the amount of primary data on nemertean pseudocnidae provided in the results section.

Additional comments

On the one hand, a commendable contribution due to its wealth of data, on the other hand, maybe due to the unusually large size of the manuscript, a more rigorous editing by the authors prior to submission would have been desirable.

Annotated reviews are not available for download in order to protect the identity of reviewers who chose to remain anonymous.

---

## Round 0.2 · accepted · Accept

Dear authors,

Many thanks for accommodating both reviewer's suggestions and for following up on the language revision.